# Time-Conditioned Dances with Simplicial Complexes: Zigzag Filtration Curve based Supra-Hodge Convolution Networks for Time-series Forecasting

**Yuzhou Chen**
Department of Computer and Information Sciences
Temple University
`yuzhou.chen@temple.edu`

**Yulia R. Gel**
Department of Mathematical Sciences
University of Texas at Dallas
National Science Foundation
`ygl@utdallas.edu`

**H. Vincent Poor**
Department of Electrical and Computer Engineering
Princeton University
`poor@princeton.edu`

## Abstract

Graph neural networks (GNNs) offer a new powerful alternative for multivariate time series forecasting, demonstrating remarkable success in a variety of spatio-temporal applications, from urban flow monitoring systems to health care informatics to financial analytics. Yet, such GNN models pre-dominantly capture only lower order interactions, that is, pairwise relations among nodes, and also largely ignore intrinsic time-conditioned information on the underlying topology of multivariate time series. To address these limitations, we propose a new time-aware GNN architecture which amplifies the power of the recently emerged simplicial neural networks with a time-conditioned topological knowledge representation in a form of zigzag persistence. That is, our new approach, Zigzag Filtration Curve based Supra-Hodge Convolution Networks (ZFC-SHCN) is built upon the two main components: (i) a new highly computationally efficient zigzag persistence curve which allows us to systematically encode time-conditioned topological information, and (ii) a new temporal multiplex graph representation module for learning higher-order network interactions. We discuss theoretical properties of the proposed time-conditioned topological knowledge representation and extensively validate the new time-aware ZFC-SHCN model in conjunction with time series forecasting on a broad range of synthetic and real-world datasets: traffic flows, COVID-19 biosurveillance, Ethereum blockchain, surface air temperature, wind energy, and vector autoregressions. Our experiments demonstrate that the ZFC-SHCN achieves the state-of-the-art performance with lower requirements on computational costs.

## 1 Introduction

Over the last few years, graph neural networks (GNNs) have emerged as a new powerful alternative to traditional statistical and machine learning models in conjunction with univariate and multivariate time series forecasting tasks [27, 4, 40, 40, 28]. Such successful applications of GNNs range from urban traffic analytics to forecasting COVID-19 hospitalizations to electrocardiogram monitoring [3, 36, 56, 10, 20]. However, most GNNs remain inherently static and do not explicitly incorporate the inherent time characteristics of the encoded knowledge [59, 42]. In turn, limitations in capturing the

36th Conference on Neural Information Processing Systems (NeurIPS 2022).

time dimension in the knowledge representation and learning mechanisms for time-evolving data results in GNNs becoming less relevant over time and, hence, requiring frequent updates.

Furthermore, GNNs tend to pre-dominantly focus only on information propagation among nodes and also be limited in their ability to describe polyadic relationships among multiple substructures of multivariate time series or multi-node interactions in dynamics graphs. However, as recently shown by [6, 21], such higher-order interactions might be the key toward better understanding of the underlying mechanisms of many real-world graph-structured phenomena. This challenge on polyadic graph interactions has been recently addressed by [24, 8, 7] who propose to model higher order substructures as simplices. Then, by borrowing the concepts of the Hodge theory, these approaches allow for generalization of the ideas of the combinatorial graph Laplacian which describes a diffusion from node to node via edges to a case of diffusion over simplices. Such Hodge Laplacian construction allows for extending the notion of convolution operation to simplicial convolution, and the resulting simplicial neural networks (SNNs) are arguably one of the frontlines in graph learning today. However, these ideas have never been yet applied in conjunction with knowledge representation and learning of time-evolving objects.

Our goal here is to bridge the emerging concept of time-aware learning with the recent notions of simplicial convolution, with a particular focus on explicitly integrating the core time-conditioned topological characteristics. In particular, we amplify the power of SNNs with a time-conditioned topological knowledge representation in a form of zigzag persistence for time-indexed data and, more specifically, its new highly computationally efficient summary, Zigzag Filtration Curve. As a result, our new approach, Zigzag Filtration Curve based Supra-Hodge Convolution Networks (ZFC-SHCN) enables us to systematically learn the most intrinsic time-conditioned information both on the underlying topology of the time-evolving data and higher-order interactions among various substructures.

Significance of our contributions can be summarized as follows:

- ZFC-SHCN is the first approach bringing the concepts of simplicial convolution and SNNs to time-aware learning.

- We propose a new highly computationally efficient summary of persistence for time-indexed data, Zigzag Filtration Curve, and derive its theoretical stability guarantees.

- We validate the utility of ZFC-SHCN in conjunction with forecasting multivariate time series from diverse application domains such as traffic networks, COVID-19 biosurveillance, surface air temperature, token prices on Ethereum blockchain, wind energy, and vector autoregressions. Our findings indicate that ZFC-SHCN delivers the state-of-the-art forecasting performance, with a significant margin and demonstrates higher computational efficiency.

## 2  Related Work

**Time-series Forecasting and Spatio-temporal Graph Convolutional Networks** Time-series forecasting is one of the core subfields in statistical sciences [15, 9]. Most recently, there have appeared a number of unconventional machine learning approaches to time-series forecasting. In particular, graph convolutional network (GCN)-based models for spatio-temporal network data have emerged as a promising forecasting tool. For instance, DCRNN [42] introduces spectral graph convolution into spatio-temporal network data prediction, which can capture spatio-temporal dependencies. STGCN [59] uses convolutional neural networks (CNNs) to model temporal correlations. Moreover, to infer hidden inter-dependencies between different traffic variables, [57, 3, 10] conduct a convolution operation in spatial dimension through adaptive adjacency matrices. Recent Z-GCNETs [20] develops a zigzag topological layer equipped with a zigzag persistence image into a GCN framework to model temporal correlations. Another promising recent direction for time series forecasting beyond GCN is a fractional-order dynamical model proposed by [27]. This approach offers an alternating scheme to determine the best estimate of the model parameters and unknown stimuli. In turn, [28] proposes a Padé approximation based exponential neural operator (Padé Exp), aiming to improve time-series forecasting with exponential operators in neural operator learning schemes. However, all of the above methods only focus on node-level representations. In contrast, in this paper, we focus on both higher-order structure representation and topological information learning.

**Topological Data Analysis for Graph Learning** Persistent homology [25, 62] is a suite of tools within topological data analysis (TDA) that provides a way for measuring topological features of shapes and functions. The extracted topological features have been recently shown to provide invaluable insights into hidden mechanisms behind the organization and functionality of graph structured data. In particular, topological features have been actively used for node classification [61, 17], link prediction [58], and graph classification [31, 32, 14, 30]. For instance, [31] is one of the first approaches to integrate topological features into neural networks for graph classification, while [14] proposes a versatile framework for learning multiple vectorizations of persistent diagrams on graphs. In turn, [61, 17, 58] apply topological features to GNNs to understand and improve the message passing between nodes. Finally, [33] proposes a topological graph layer with learnable filtration functions for graph and node classification tasks, while [13] advances the ideas of multipersistence to graph learning.

**Zigzag Persistent Homology** Despite its promise, regular persistent homology does not explicitly model the geometric and topological information from a sequence of topological spaces. To address this limitation, a generalization of ordinary persistence, i.e., zigzag persistent homology, based on the theory of quiver representation, has been proposed by [12]. Zigzag persistence allows us to systematically describe how the homology changes over a sequence of spaces. Despite its high potential, especially in conjunction with analysis of time-evolving data, zigzag persistence still remains largely a theoretical concept, and its applications are yet scarce. The recent results for time-dependent data studies include, for example, zigzag-based clustering [37], bifurcation analysis of dynamic systems [55], and time series forecasting [20]. The memory and computational efficiency of zigzag persistence is one of the daunting challenges. Inspired by [44], we propose a novel highly computationally efficient representation of zigzag persistence for learning time-evolving data, that is, zigzag filtration curve.

**Simplicial Neural Networks** Modeling higher-order interactions on graphs is an emerging direction in graph representation learning. While the role of higher-order structures for graph learning has been documented for a number of years [1, 34] and involves such diverse applications as graph signal processing in image recognition [23], dynamics of disease transmission and biological networks, integration of higher-order graph substructures into deep learning on graphs has emerged only in 2020. As shown by [6, 50], higher-order network structures can be leveraged to boost graph learning performance. Indeed, several recent approaches [24, 49, 8, 18] propose to leverage simplicial information to perform neural networks on graphs. However, neither of these Simplicial Neural Networks (SNNs) are integrated with a topology-based graph convolution layer allowing us to learn both time-aware persistent topological features and simplicial geometry of graphs. In this paper, we propose ZFC-SHCN to address this limitation.

## 3 Time-Aware Topological Learning with Zigzag Curves

**Spatio-temporal Graph Construction** A spatio-temporal graph is a collection of snapshots at different time steps, denoted by $\mathcal{G} = \{\mathcal{G}_1, \mathcal{G}_2, \cdots, \mathcal{G}_{\mathcal{T}}\}$, where $\mathcal{T}$ is the maximum timestamp. Here $\mathcal{G}_t = (\mathcal{V}_t, \mathcal{E}_t, \boldsymbol{\mathcal{A}}_t, \boldsymbol{X}_t)$ is the graph observed at time step $t \in [1, \mathcal{T}]$, where $\mathcal{V}_t$ is a finite set of $|\mathcal{V}| = N$ nodes, $\mathcal{E}_t$ is a set of edges, $\boldsymbol{\mathcal{A}}_t \in \mathbb{R}^{N \times N}$ is the adjacency matrix, and $\boldsymbol{X}_t \in \mathbb{R}^{N \times d}$ is the node feature matrix. Specifically, each row of $\boldsymbol{X}_t$ is a $d$-dimensional feature vector of the corresponding node. For sake of notations, wherever applicable below, we omit the subscript $t$ and denote graph $\mathcal{G}_t$ at time $t$ as $\mathcal{G}$.

**Background on Ordinary Persistence** Tools of ordinary persistence, or persistent homology (PH), allow us to study salient data shape patterns along various dimensions. By shape here we broadly understand data properties that are invariant under continuous transformations, that is, transformations that do not alter "holes" in the data, for example, bending, twisting, and stretching. The key idea is to choose some suitable scale parameter $\nu$ and then to study a graph $\mathcal{G}$ not as a single object but as a nested sequence of graphs, or *graph filtration* $\mathcal{G}_1 \subseteq \ldots \subseteq \mathcal{G}_n = \mathcal{G}$, which is induced by monotonic changes of scale $\nu$. For example, if $\mathcal{G}$ is an edge-weighted graph $(\mathcal{V}, \mathcal{E}, w)$ with weight function $w : \mathcal{E} \mapsto \mathbb{R}$, then for each $\nu_j$, $j = 1, \ldots, n$, we set $\mathcal{G}_{\leq \nu_j} = (\mathcal{V}, \mathcal{E}, w^{-1}(-\infty, \nu_j])$, yielding the induced edge-weighted filtration. We can also consider only induced subgraphs of $\mathcal{G}$ with maximal degree of $\nu_j$ for each $j = 1, \ldots, n$, resulting in the degree sublevel set filtration. (For more discussion on graph filtrations see [30].) Armed with this construction, we can track which shape patterns, for example, independent components, loops, and voids, emerge as the scale $\nu$ varies. To make the

process of pattern counting more systematic and efficient, we build an abstract simplicial complex $\mathscr{K}(\mathcal{G}_j)$ on each $\mathcal{G}_j$. We also record complex indices $j_b$ (birth) and $j_d$ (death) at which we first or last observe each shape feature. Topological features with longer lifespans are said to persist and are likelier to yield important information on the structural organization of $\mathcal{G}$.

**Learning Shapes of Time-Conditioned Data with Zigzag Persistence** This construction enables us to extract the key topological descriptors from a single graph $\mathcal{G}$. However, in our case, we observe not a single graph but a sequence of time-evolving graphs $\{\mathcal{G}^1, \ldots, \mathcal{G}^{\mathcal{T}}\}$. How can we track shape signatures which are not just individualistic for each time stamp but characterize intrinsic properties of the observed object over time? One approach to how we can bring PH tools to analysis of time-conditioned objects is zigzag persistence. Based on the theory of quiver representations, zigzag persistence generalizes ordinary persistence to track characteristics of graphs (or other topological spaces) with inclusions going in different directions [12, 11]. In particular, given a time-indexed sequence of graphs $\{\mathcal{G}^1, \ldots, \mathcal{G}^{\mathcal{T}}\}$, we first form a set of graph inclusions over time

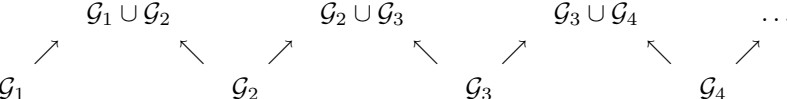

and then assess the compatibility of persistent topological features across unions of graphs. That is, we record indices at which topological features (dis)appear, for some given scale $\nu_*$. If for a given $\nu_*$ topological feature $\rho$ (i.e., $p$-dimensional hole, $0 \le p \le \mathcal{K}$, where $\mathcal{K}$ is the dimension of the simplicial complex $\mathbb{K}(\mathcal{G})$) is first recorded in $\mathbb{K}(\mathcal{G}^j)$, we say that the feature's birth is $j$, and if $\rho$ first appears in $\mathbb{K}(\mathcal{G}^j \cup \mathcal{G}^{j+1})$, we record its birth as $j + 1/2$. In turn, if $\rho$ is last seen in $\mathbb{K}(\mathcal{G}^j)$, we record its death as $j$, while if it is last seen in $\mathbb{K}(\mathcal{G}^j \cup \mathcal{G}^{j+1})$, we say that its death is at $j + 1/2$. Let $\mathfrak{J}$ be the set of all observed topological features for a given $\nu_*$. Collecting then births and deaths over $\mathfrak{J}$, we summarize all extracted information as a multiset $\mathcal{D}_{\nu_*} = \{(b_\rho, d_\rho) \in \mathbb{R}^2 | b_\rho < d_\rho, \rho \in \mathfrak{J}\}$, called a zigzag persistent diagram (ZPD) (where $b_\rho$ and $d_\rho$ are the birth and death of the topological feature $\rho$ respectively).

This makes zigzag persistence particularly attractive for the analysis of dynamic objects which are naturally indexed by time. However, the idea of zigzag persistence is applicable far beyond learning time-evolving objects. Nevertheless, zigzag persistence still remains largely a theoretical concept, with yet only a handful of applications, and one of the roadblocks hindering a broader proliferation of zigzag-based methods in practice is their computational costs. Here we take a step toward bringing a more computationally efficient summary of zigzag persistence to real-world applications.

**Time-Aware Zigzag Filtration Curves** Consider a sequence of time intervals associated with a zigzag filtration over a time period $[t_1, t_{\mathcal{N}}]$

$$\left(t_1, t_1 + \frac{1}{2}\right), \left(t_1 + \frac{1}{2}, t_2\right), \left(t_2, t_2 + \frac{1}{2}\right), \ldots, \left(t_{\mathcal{N}-1} + \frac{1}{2}, t_{\mathcal{N}}\right).$$

Let $\mathrm{DgmZZ}_{\nu_*}$ be the resulting ZPD for a given $\nu_*$ and $\mathcal{M}$ be the number of off-diagonal topological features in ZPD, i.e., $\mathrm{DgmZZ}_{\nu_*}$. Inspired by the recent results on stabilized Betti sequences by [35] and filtration curves by [44] for ordinary persistence, we propose a new simple and computationally efficient summary of zigzag persistence, called a *Zigzag Filtration Curve*.

**Definition 3.1** (Zigzag Filtration Curve (ZFC)). The zigzag filtration curve evaluated at $\Delta t_i^- = (t_{i-1} + \frac{1}{2}, t_i)$, $i = \{1, 2, \ldots, \mathcal{N}\}$, for a given $\nu_*$, is defined as

$$\mathrm{ZFC}_{\nu_*}^p(\Delta t_i^-) = \sum_{j=1}^{\mathcal{M}} \xi_i(t_{b_j}, t_{d_j}) \omega_i,$$

where $(t_{b_j}, t_{d_j}) \in \mathbb{R}^2$ is a vector containing the birth and death of the $j$-th off-diagonal $p$-dimensional topological feature in $\mathrm{DgmZZ}_{\nu_*}$ (as such, $t_{b_j} < t_{d_j}$), $j = \{1, 2, \ldots, \mathcal{M}\}, 0 \le p \le \mathcal{K}; \xi_i : \mathbb{R}^2 \mapsto \mathbb{R}$ is some suitable Lipschitz continuous function with Lipschitz constant $L_i$, for example, a Gaussian density; and $\omega_i > 0, i = \{1, 2, \ldots, \mathcal{N}\}$ are weights such that $\sum_i \omega_i = 1$. Zigzag filtration curve at $\Delta t_i^+ = (t_i, t_i + \frac{1}{2})$ is defined analogously. (For the sake of notational simplicity, wherever applicable in the further exposition we suppress the index $p$ in ZFC.)

Motivated by [35], here as the Lipschitz continuous function $\xi_i$ for intervals $\Delta t_i^-$, we use a Gaussian density $f$ with mean $(t_{i-1} + 1/2, t_i)$, while for intervals $\Delta t_i^+$, we set the mean of $f$ to $(t_i, t_i + 1/2)$,

$i = 1, 2, \ldots \mathcal{N}$. For both $\Delta t_i^-$ and $\Delta t_i^+$, we choose the $2 \times 2$-variance-covariance matrix $\Sigma$ to be the identity matrix. (See Appendix **??** for more discussion on sensitivity analysis.) Another suitable choice of $\xi$ is the arctan function.

As we show below, the proposed ZPC also enjoys important theoretical stability guarantees in terms of Wasserstein-1 distance.

**Proposition 3.2** (Stability of Zigzag Filtration Curve). *Let $DgmZZ_{\nu_*}$ be a zigzag persistence diagram and $DgmZZ'_{c_*}$ be its perturbed copy such that $\mathcal{W}_1\big(DgmZZ_{\nu_*}, DgmZZ'_{\nu_*}\big) < \epsilon$, where $\mathcal{W}_1$ is Wasserstein-1 distance. Then, ZFC is stable with respect to Wasserstein-1 distance.*

In practice topological features of various dimensions $p$, $p = 0, 1, \ldots, \mathcal{K}$, may play different roles in the learning task performance, and these roles are not known a-priori. Hence, to harness time-conditioned information encoded in ZFC corresponding to different dimensions $p$, we propose Multi-Zigzag Filtration Curves (M-ZFCs) M-ZFCs$_{\nu_*} \in \mathbb{R}^{K \times \frac{\mathcal{N}-1}{2}}$ by stacking $\text{ZFC}^0, \text{ZFC}^1, \ldots, \text{ZFC}^{\mathcal{K}}$. Figure **??** in Appendix **??** shows the both 0- and 1-dimensional ZFCs obtained from the proposed ZFC. In the following section, we demonstrate how ZFC can be integrated into neural network architectures for graph learning tasks.

# 4 Zigzag Filtration Curve Based Supra-Hodge Convolution Networks

Given a graph $\mathcal{G}$ and its historical $\omega$ step graph signals $\boldsymbol{X}^\omega = \{\boldsymbol{X}_{t-\omega+1}, \ldots, \boldsymbol{X}_t\} \in \mathbb{R}^{\omega \times N \times F}$ ($F$ is the node feature dimensionality), the time-series forecasting problem is to learn a mapping function $f$ that maps the historical data $\{\boldsymbol{X}_{t-\omega+1}, \ldots, \boldsymbol{X}_t\}$ into the next $h$ step data $\{\boldsymbol{X}_{t+1}, \ldots, \boldsymbol{X}_{t+h}\}$. The mapping relation is represented as follows: $\{\boldsymbol{X}_{t-\omega+1}, \ldots, \boldsymbol{X}_t\} \xrightarrow{f} \{\boldsymbol{X}_{t+1}, \ldots, \boldsymbol{X}_{t+h}\}$.

## 4.1 Graph convolution in the spatial dimension

Given the node embedding dictionary $\boldsymbol{W}^\phi = (w_1^\phi, w_2^\phi, \ldots, w_N^\phi) \in \mathbb{R}^{N \times d_c}$ (where $x_u^\phi \in \mathbb{R}^{d_c}$ and $d_c$ is the dimension of node embedding), we aim to seek a non-negative function $\boldsymbol{S}_{u,v} = \mathscr{G}(w_u^\phi, w_v^\phi)$ which represents the pairwise similarity between any two nodes $u$ and $v$. Concretely, the multiplication between $\boldsymbol{W}^\phi$ and $(\boldsymbol{W}^\phi)^\top$ can (i) give a sum pooling of second-order features from the outer product of all the embedding vector pairs $(w_u^\phi, w_v^\phi)$ and (ii) infer the hidden spatial dependencies of nodes

$$\boldsymbol{S}_{uv} = \mathscr{G}(w_u^\phi, w_v^\phi) = \frac{\exp\big(\text{ReLU}(w_u^\phi(w_v^\phi)^\top)\big)}{\sum_{u=1}^N \exp\big(\text{ReLU}(w_u^\phi(w_v^\phi)^\top)\big)},$$

where $\text{ReLU}(\cdot) = \max(0, \cdot)$ is a nonlinear activation function, which is used to eliminate weak connections proactively, and the role of the softmax function is applied to normalize the learned graph $\boldsymbol{S}$. Inspired by the recent advancements in random walk-based graph embedding learning [47, 26], we make a graph convolution in spatial dimension, feeding a power series of the learned graph $\boldsymbol{S}$ with varying random walk steps $\{1, 2, \cdots, r\}$ ($r \in \mathbb{Z}_+$), as follows:

$$\boldsymbol{H}_{t,\text{GC}}^{(\ell+1)} = \sigma(\text{Stack}(\boldsymbol{I}, \boldsymbol{S}, \cdots, \boldsymbol{S}^r)\boldsymbol{H}_{t,\text{GC}}^{(\ell)}\boldsymbol{\Theta}_{\text{GC}}^{(\ell)}), \tag{1}$$

where $\sigma(\cdot)$ stands for a nonlinear activation function, $\text{Stack}(\cdot)$ is the function which stacks $r$ powered learned graphs, $\boldsymbol{H}_{t,\text{GC}}^{(\ell)}$ and $\boldsymbol{H}_{t,\text{GC}}^{(\ell+1)}$ are the input and output activations for layer $\ell$ (where $\boldsymbol{H}_{t,\text{GC}}^{(0)} = \boldsymbol{X}_t \in \mathbb{R}^{N \times F}$), and $\boldsymbol{\Theta}_{\text{GC}}^{(\ell)} \in \mathbb{R}^{d_\ell^{\text{GC}} \times d_{\ell+1}^{\text{GC}}}$ is the $\ell$-th layer's trainable weights. Next, we introduce representation learning of the higher-order graph (sub)structures using the supra-Hodge Laplacian which allows us to systematically leverage the underlying topological information.

## 4.2 Supra-Hodge convolution in temporal dimension

Time-evolving data such as multivariate time series, spatio-temporal processes, and dynamic networks, often exhibit a highly complex dependency among its substructures that goes far beyond what can be described by dyadic (or pairwise) interactions among nodes. Instead, such higher-order polyadic interactions can be systematically addressed using the Hodge theory. In particular, the discrete Hodge theory allows us to generalize the notion of a standard combinatorial graph Laplacian which describes diffusion on graph $\mathcal{G}$ from node to node via edges to diffusion over higher-order

substructures of $\mathcal{G}$ [43, 6]. In turn, higher-order substructures can be modeled as $k$-simplices of $\mathcal{G}$. (See Appendix **??** for background information on Hodge Laplacians.) Convolutional architectures on simplicial complexes based on the associated concepts of the Hodge theory have emerged as a recent direction in graph neural networks but have not yet been applied to learning time-evolving data. Our goal here is to introduce the notion of simplicial convolution and the ideas of Hodge-Laplacians to time-aware learning.

In particular, to capture time-conditioned higher-order interactions on $\mathcal{G}$ and to describe diffusion of information over simplices along the temporal dimension, we build a supra-Hodge convolution operation, based on the multiplex network representation learning. (In the following for simplicity, notation without sub/superscript $k$ stands for node-level quantities and in our experiments we always consider $k \in \mathbb{Z}_+$). First, given the historical spatio-temporal network series $\mathcal{G}_{t-\omega+1:t} = \{\mathcal{G}_{t-\omega+1}, \mathcal{G}_{t-\omega+2}, \ldots, \mathcal{G}_t\}$, we consider a directed connected node-aligned multiplex network, which is made up of $\omega$ layers with $N$ nodes on each layer. That is, the adjacency matrix $\mathcal{A}^\alpha = \{a_{uv}^\alpha\}_{N \times N}$ (where $\alpha \in \{t - \omega + 1, \ldots, t\}$) defines the intra-connection between nodes $u$ and $v$ in layer $\alpha$ and a distance matrix $\mathcal{D}^{\alpha\beta} = \{d_{uu}^{\alpha\beta}\}_{N \times N}$ quantifies the transition probability of moving from node $u$ of layer $\alpha$ to node $u$ of layer $\beta$. (Here $\beta > \alpha$, since we consider information diffusion procedures only along the temporal dimension). Next, based on the discrete Hodge theory, we propose a new Hodge $k$-Laplacian for multiplex graphs, called the *supra-Hodge $k$-Laplacian* $\mathcal{L}_k^{Sup} \in \mathbb{R}^{\phi_k \omega \times \phi_k \omega}$

$$\mathcal{L}_k^{\text{Sup}} = \begin{pmatrix} (\mathcal{L}_k^{11})^r & D_{k+1}^{12} & \cdots & D_{k+1}^{1\omega} \\ \mathbf{0} & (\mathcal{L}_k^{22})^r & \cdots & D_{k+1}^{2\omega} \\ \vdots & \vdots & \ddots & \vdots \\ \mathbf{0} & \mathbf{0} & \cdots & (\mathcal{L}_k^{\omega\omega})^r \end{pmatrix}, \tag{2}$$

where $\mathcal{L}_k^{\alpha\alpha}$ is the Hodge $k$-Laplacian in layer $\alpha$, $D_{k+1}$ is the diagonal matrix of degrees of each $k$-simplex, i.e., $D_{k+1} = \max(\text{diag}(|B_{k+1}|\mathbf{1}, I))$ and $B_{k+1}$ is the $k$-simplex-to-$(k+1)$-simplex incidence matrix, and the $r$-th power of $\mathcal{L}_k^{\alpha\alpha}$ represents $r$-step random walk on the Hodge $k$-Laplacian of layer $\alpha$ which will allow every $k$-simplex to accumulate information from its neighbors. Hence, when $k = 1$, we can infer the spatial dependencies between each pair of edges and capture meaningful edge information in both spatial and temporal dimensions – through the lens of the supra-Hodge 1-Laplacian. For instance, in molecule networks, each node represents an atom and each edge is a bond connecting two atoms; the bond (i.e., edge) features include bond type, ring status, and molecular charge which are closely related to atom (i.e., node) features (such as atomic total and partial charges).

Since the goal of the forecasting task is to predict node (i.e., 0-simplex) attribute(s) in the next few time steps, we propose a novel diffusion supra-Hodge convolution on the sliding window $\mathcal{G}_{t-\omega+1:t}$. We then update nodes' representations by transforming the multiplex $k$-simplex embedding to nodes via incidence matrices

$$H_{t,k,\text{SH}}^{(\ell+1)} = \sigma(\mathcal{L}_k^{\text{Sup}} H_{t,k,\text{SH}}^{(\ell)} \Theta_{k,\text{SH}}^{(\ell)}), \tag{3}$$

$$H_{t,\text{SH}}^{(\ell+1)} = (B_1^\top \cdots B_k^\top) H_{t,k,\text{SH}}^{(\ell+1)}, \tag{4}$$

where (i) in Equation 3: $\Theta_{k,\text{SH}}^{(\ell)} \in \mathbb{R}^{d_{k;\ell}^{\text{SH}} \times d_{k;\ell+1}^{\text{SH}}}$ is a learnable filter matrix for layer $\ell$ (here $d_{k;\ell}^{\text{SH}}$ and $d_{k;\ell+1}^{\text{SH}}$ are the intermediate and output dimensions to the $\ell$-th layer), $H_{t,k,\text{SH}}^{(\ell)}$ and $H_{t,k,\text{SH}}^{(\ell+1)}$ are the input and output activations for layer $\ell$ (where $H_{t,k,\text{SH}}^{(0)} = \bar{X}_{k;t-\omega+1:t} \in \mathbb{R}^{\phi_k \omega \times d_k^{\text{in}}}$ and the historical $k$-simplex features of the spatio-temporal networks $X_{k;t-\omega+1:t} = \{X_{k;t-\omega+1}, X_{k;t-\omega+2}, \ldots, X_{k;t}\} \in \mathbb{R}^{\phi_k \times \omega \times d_k^{\text{in}}}$ is reshaped as a matrix $\bar{X}_{k;t-\omega+1:t}$ with shape $\phi_k \omega \times d_k^{\text{in}}$) and (ii) in Equation 4: we transform the $k$-simplex embedding $H_{t,k,\text{SH}}^{(\ell+1)}$ to node embedding $H_{t,\text{SH}}^{(\ell+1)} \in \mathbb{R}^{N \times d_{k;\ell+1}^{\text{SH}}}$ through incidence matrices.

### 4.3 ZFC convolution: a bridge between spatial and time dimensions

Armed with the representation learning of graph (sub)structures at each timestamp, we now discuss the ZFC convolution which allows us to preserve and propagate both spatial and time-aware topological information simultaneously. The intuition behind ZFC convolution is that it learns a strong connection between two dimensions via two 1D convolution layers, i.e., time-wise and node-wise. ZFC

convolution consists of three key components: (i) a linear embedding on M-ZFCs, which can learn the importance of time-aware topological features for each node to form a time-dimension-specific node embedding; (ii) a time-wise 1D convolution layer, where it gathers time-aware topological features from the entire space into a compact set; (iii) a node-wise 1D convolution layer, which can capture relations between different nodes. The resulted ZFC convolution operation over a M-ZFCs$^\omega$ is defined as

$$\boldsymbol{H}_{t,\text{M-ZFC}} = \mathscr{F}_\theta(\mathscr{F}_\psi(\boldsymbol{\Theta}_{\text{M-ZFC}}\text{M-ZFCs}^\omega)^\top)^\top, \tag{5}$$

where $\omega$ is the size of the window for sequence learning, M-ZFCs$^\omega$ denotes the M-ZFCs feature extracted from the time window with size $\omega$, $\boldsymbol{\Theta}_{\text{M-ZFC}} \in \mathbb{R}^{N \times d_q}$ is a weight matrix to be learned, $\mathscr{F}_\theta$ and $\mathscr{F}_\psi$ are 1D convolutional layers, and $\boldsymbol{H}_{t,\text{M-ZFC}} \in \mathbb{R}^{N \times d_{\text{out}}^{\text{M-ZFC}}}$ is the $d_{\text{out}}^{\text{M-ZFC}}$-dimensional output. We then combine the embeddings from graph convolution, M-ZFCs convolution, and supra-Hodge convolution to get the final embedding $\boldsymbol{H}_{t,\text{out}}^{(\ell+1)}$

$$\boldsymbol{H}_{t,\text{out}}^{(\ell+1)} = [\boldsymbol{H}_{t,\text{GC}}^{(\ell+1)}, \boldsymbol{H}_{t,\text{M-ZFCs}}, \boldsymbol{H}_{t,\text{SH}}^{(\ell+1)}], \tag{6}$$

where $[\cdot, \cdot, \cdot]$ denotes the concatenation of the outputs from three convolution operations, and $\boldsymbol{H}_{t,\text{out}}^{(\ell+1)} \in \mathbb{R}^{N \times d_{\text{out}}}$ (where $d_{\text{out}} = d_{\ell+1}^{\text{GC}} + d_{\text{out}}^{\text{ZFC}} + d_{\ell+1}^{\text{SH}}$).

## 4.4 Gate Recurrent Unit with ZFC-SHCN

To describe the complex spatio-temporal dependencies among time series and assess a hidden state of nodes at a future timestamp, we feed the final embedding $\boldsymbol{H}_{t,\text{out}}^{(\ell+1)}$ into Gated Recurrent Units (GRUs). Formally, we set the forward propagation equations of the GRUs as

$$\boldsymbol{\Re}_t = \eta\left(\boldsymbol{W}_\Re\left[\boldsymbol{\Psi}_{t-1}, \boldsymbol{H}_{t,\text{out}}^{(\ell+1)}\right] + \boldsymbol{b}_\Re\right), \qquad \boldsymbol{\Im}_t = \eta\left(\boldsymbol{W}_\Im\left[\boldsymbol{\Psi}_{t-1}, \boldsymbol{H}_{t,\text{out}}^{(\ell+1)}\right] + \boldsymbol{b}_\Im\right),$$

$$\boldsymbol{\Psi}_t = \tanh\left(\boldsymbol{W}_\Psi\left[\boldsymbol{\Im}_t \odot \boldsymbol{\Psi}_{t-1}, \quad \boldsymbol{H}_{t,\text{out}}^{(\ell+1)}\right] + \boldsymbol{b}_\Psi\right), \quad \tilde{\boldsymbol{\Psi}}_t = \boldsymbol{\Re}_i \odot \boldsymbol{\Psi}_{t-1} + (1 - \boldsymbol{\Re}_t) \odot \boldsymbol{\Psi}_t,$$

where $\eta(\cdot)$ is an activation function (e.g., ReLU, LeakyReLU), $\odot$ is the elementwise product, $\boldsymbol{\Re}_t$ is the update gate and $\boldsymbol{\Im}_i$ is the reset gate. Here $\boldsymbol{b}_\Re$, $\boldsymbol{b}_\Im$, $\boldsymbol{b}_\Psi$, $\boldsymbol{W}_\Re$, $\boldsymbol{W}_\Im$, and $\boldsymbol{W}_\Psi$ are learnable parameters, while $\left[\boldsymbol{\Psi}_{t-1}, \boldsymbol{H}_{t,\text{out}}^{(\ell+1)}\right]$ and $\boldsymbol{\Psi}_t$ are the input and output of GRU model, respectively. We then obtain $\tilde{\boldsymbol{\Psi}}_t$ which contains both the spatio-temporal and time-aware information.

# 5 Experiments

## 5.1 Datasets

We validate our ZFC-SHCN model on six diverse data types: (i) COVID-19 datasets [51]: CA, PA, and TX represent the number of COVID-19 hospitalizations in California (CA), Pennsylvania (PA), and Texas (TX) respectively; (ii) traffic datasets [16]: PeMSD4 and PeMSD8 are two real-time traffic datasets from California; (iii) synthetic multivariate time-series (MTS) datasets based on vector autoregression (VAR) [29, 45] (where the VAR model is a generalization of the univariate AR process with more than one time-evolving component); (iv) daily surface air temperature in CA, PA, and TX over 02/01/2020–12/31/2020; (v) Bytom token prices of Ethereum blockchain over 07/27/2017–05/07/2018 [41, 53]; and (vi) wind speed data of 57 stations on the East Coast. The results on (i)–(iii) are presented in the main body, and the analysis of (iv) and (v) is in Appendix **??** and **??**. The detailed description of each dataset is in Appendix **??**. We also report results on the wind speed dataset in Appendix **??**.

## 5.2 Baselines

We compare our proposed ZFC-SHCN with 14 types of state-of-the-art baselines (SOAs), including FC-LSTM [54], SFM [60], N-BEATS [46], DCRNN [42], LSTNet [38], STGCN [59], TCN [4], DeepState [48], GraphWaveNet [57], DeepGLO [52], LRGCN [39] AGCRN [3], StemGNN [10], and Z-GCNETs [20].

## 5.3 Experimental settings

We implement ZFC-SHCN within a Pytorch framework on NVIDIA GeForce RTX 3090 GPU. We optimize all the models using an Adam optimizer for a maximum of 200 epochs. The learning rate is searched in $\{0.001, 0.003, 0.005, 0.01, 0.05\}$ and the embedding dimension is searched in $\{1, 2, 3, 5, 10\}$. Our ZFC-SHCN is trained with batch sizes of 64 and 8 on PeMSD4 and PeMSD8, respectively. On both COVID-19 and surface air temperature datasets (i.e., CA, PA, and TX), we set the batch size to be 8. We train two 1D convolutional layers for ZFC representation learning with the same hidden layer dimension $nhid$ where $nhid \in \{8, 16, 32, 64, 128\}$. For PeMSD4 and PeMSD8, we consider the window size $\omega = 12$ and the horizon $h = 3$; for both COVID-19 and surface air temperature datasets, we consider a window size $\omega = 5$ and horizon $h = 15$; for two simulated VAR datasets $\text{VAR}_{T_1}$ and $\text{VAR}_{T_2}$, we set the window size as $\omega = 10$ and horizon as $h = 5$, and set the batch size as 8; for Bytom, we consider the window size $\omega = 7$ and horizon $h = 7$, and set the batch size as 8; for the wind speed dataset, we consider the window size $\omega = 12$ and horizon $h = 12$, and set the batch size as 8. All models are evaluated in terms of Mean Absolute Error (MAE), Root Mean Square Error (RMSE), and Mean Absolute Percentage Error (MAPE). The best results are shown in **bold** font and the results shown with *dotted underlines* are the second-best results. We also perform a one-sided two-sample $t$-test between the best result and the best performance achieved by the runner-up, where *, **, *** denote $p$-value $< 0.1, 0.05, 0.01$ (i.e., denote significant, statistically significant, and highly statistically significant results, respectively. Code is available at `https://github.com/zfcshcn/ZFC-SHCN.git`.

## 5.4 Experimental results

**Real datasets** The experimental results on PeMSD4 and PeMSD8 traffic data are reported in Table 2. As Table 2 shows, ZFC-SHCN achieves the best MAE, RMSE, and MAPE compared with SOAs on both PeMSD4 and PeMSD8. Compared to the RNN-based methods such as FC-LSTM, SFM, N-BEATS, LSTNet, and TCN, ZFC-SHCN achieves relative gains in RMSE over the runner-ups, ranging from 17.68% to 65.41% for both PeMSD4 and PeMSD8. In turn, DCRNN, STGCN, GraphWaveNet, AGCRN, and StemGNN only focus on learning node-level representations. Compared to them, ZFC-SHCN captures interactions and encodes higher-order structure correlations beyond pairwise relations among nodes and yields a relative gain from 2.06% to 5.63% in RMSE on the traffic datasets. In addition, we compare ZFC-SHCN with the method based on the zigzag persistence image, i.e., Z-GCNETs, and find that ZFC-SHCN outperforms Z-GCNETs by 1.75% on PeMSD4 and 5.36% on PeMSD8 in terms of RMSE. Table 3 presents COVID-19 hospitalization prediction results (RMSE) in CA, PA, and TX, and we observe the following findings. First, our proposed ZFC-SHCN achieves state-of-the-art performance on all three datasets. For instance, ZFC-SHCN yields 3.61%, 1.47%, 65.55% relative gains in RMSE over the runner-ups (including both GCN-based and zigzag persistence image-based methods) on three biosurveillance datasets. These results indicate that the ZFC mechanism and higher-order representation learning module play significant roles in capturing both topological information and higher-order structures. Second, as shown in Figure **??** in Appendix **??**, we find that, compared to the runner-up (i.e., Z-GCNETs), the predicted value of COVID-19 hospitalizations is more consistent with the ground-truth. Finally, Tables **??** and **??** in Appendix **??** present the overall prediction performances of ZFC-SHCN and representative baselines on surface air temperature and Ethereum blockchain datasets. We find that our proposed ZFC-SHCN consistently outperforms all baselines with either a significant or (highly) statistically significant margin across all data, except surface air temperature in TX, where ZFC-SHCN still yields the best performance across all models.

Table 1: Forecasting performance (RMSE) of ZFC-SHCN and top three baselines on synthetic time series following vector autoregressions (VAR).

| Model | $\text{VAR}_{T_1}$ | $\text{VAR}_{T_2}$ |
|---|---|---|
| AGCRN [3] | 0.50±0.03 | 0.44±0.02 |
| StemGNN [10] | 0.51±0.02 | 0.42±0.02 |
| Z-GCNETs [20] | 0.49±0.02 | 0.44±0.01 |
| **ZFC-SHCN (ours)** | ***0.45±0.01** | ***0.38±0.01** |

**Synthetic datasets** The evaluation results on two VAR datasets are summarized in Table 1. Compared to the three strongest baselines (i.e., AGCRN, StemGNN, and Z-GCNETs), we observe that our proposed ZFC-SHCN consistently yields the best performance for all synthetic datasets. More precisely, ZFC-SHCN outperforms the runner-ups from 8.89% to 10.52% for $\text{VAR}_{T_1}$ and $\text{VAR}_{T_2}$. Furthermore, to assess the time-wise and high network interactions, we use the global clustering coefficient (GCC) and Euler-Poincaré characteristic (EPC) as measures of higher order substructures [5]. We find that

Table 2: Performance comparison of all methods on PEMSD4 and PEMSD8 traffic data.

| Model | PeMSD4 | | | PeMSD8 | | |
|---|---|---|---|---|---|---|
| | MAE | RMSE | MAPE (%) | MAE | RMSE | MAPE (%) |
| FC-LSTM [54] | 27.14 | 41.59 | 18.20 | 22.20 | 34.06 | 14.20 |
| SFM [60] | 24.36 | 37.10 | 17.20 | 16.01 | 27.41 | 10.40 |
| N-BEATS [46] | 25.56 | 39.90 | 17.18 | 19.48 | 28.32 | 13.50 |
| DCRNN [42] | 24.70 | 38.12 | 17.12 | 17.86 | 27.83 | 11.45 |
| LSTNet [38] | 24.04 | 37.38 | 17.01 | 20.26 | 31.96 | 11.30 |
| STGCN [59] | 22.70 | 35.50 | 14.59 | 18.02 | 27.83 | 11.40 |
| TCN [4] | 26.31 | 36.11 | 15.62 | 15.93 | 25.69 | 16.50 |
| DeepState [48] | 26.50 | 33.00 | 15.40 | 19.34 | 27.18 | 16.00 |
| GraphWaveNet [57] | 26.85 | 39.70 | 17.29 | 19.13 | 28.16 | 12.68 |
| DeepGLO [52] | 25.45 | 35.90 | 12.20 | 15.12 | 25.22 | 13.20 |
| AGCRN [3] | 17.78 | 29.17 | 11.79 | 14.59 | 23.06 | 9.29 |
| StemGNN [10] | 20.20 | 31.83 | 12.00 | 15.83 | 24.93 | 9.26 |
| Z-GCNETs [20] | 18.05 | 29.08 | 11.79 | 14.52 | 23.00 | 9.28 |
| **ZFC-SHCN (ours)** | ***17.58** | ***28.58** | ***11.68** | ***13.86** | ***21.83** | ***8.92** |

Table 3: Performance on COVID-19 hospitalizations in CA, PA, and TX.

| Model | CA | | | PA | | | TX | | |
|---|---|---|---|---|---|---|---|---|---|
| | MAE | RMSE | MAPE (%) | MAE | RMSE | MAPE (%) | MAE | RMSE | MAPE (%) |
| FC-LSTM [54] | 167.86±3.25 | 502.29±4.16 | 90.71±7.17 | 47.60±2.36 | 108.74±2.28 | 69.37±2.38 | 20.75±0.50 | 71.66±2.15 | 89.16±8.96 |
| SFM [60] | 103.98±1.46 | 475.66±3.23 | 66.91±5.70 | 45.91±2.42 | 106.12±1.87 | 70.15±5.60 | 17.49±0.58 | 70.18±1.91 | 85.85±8.25 |
| N-BEATS [46] | 105.99±1.51 | 476.10±2.89 | 67.55±5.16 | 46.91±2.82 | 105.86±1.95 | 15.82±4.39 | 21.25±0.56 | 69.26±1.61 | 85.11±8.77 |
| DCRNN [42] | 107.20±1.00 | 492.10±2.96 | 69.83±5.57 | 47.49±2.13 | 107.21±1.63 | 67.15±3.94 | 17.15±0.55 | 70.47±2.28 | 88.95±8.01 |
| LSTNet [38] | 105.63±1.82 | 480.61±3.91 | 67.72±2.17 | 46.01±2.70 | 105.67±1.76 | 72.33±4.21 | 13.06±0.53 | 72.93±2.20 | 84.23±9.01 |
| STGCN [59] | 102.88±1.11 | 470.52±3.06 | 69.73±5.69 | 52.69±2.40 | 106.78±1.87 | 69.36±4.59 | 12.69±0.49 | 63.26±2.25 | 60.47±8.37 |
| TCN [4] | 110.82±1.35 | 492.82±3.54 | 70.00±6.92 | 49.80±2.36 | 105.07±1.90 | 69.86±4.91 | 13.70±0.40 | 67.08±1.89 | 65.23±8.61 |
| DeepState [48] | 100.57±1.58 | 469.15±3.71 | 68.24±5.79 | 48.46±2.70 | 107.61±2.31 | 67.69±4.21 | 13.24±0.46 | 59.47±1.93 | 62.38±8.60 |
| GraphWaveNet [57] | 89.64±1.80 | 394.83±3.35 | 67.61±5.36 | 48.08±2.18 | 109.41±1.91 | 69.39±4.27 | 12.66±0.56 | 58.98±1.22 | 60.33±8.87 |
| DeepGLO [52] | 95.58±1.94 | 455.80±3.18 | 67.35±5.99 | 47.66±2.57 | 103.74±2.07 | 68.71±4.09 | 12.02±0.60 | 53.09±1.97 | 62.88±8.83 |
| AGCRN [3] | 87.24±1.77 | 448.27±2.78 | 66.30±5.57 | 44.69±2.49 | 103.79±3.08 | 63.45±4.00 | 10.93±0.49 | 52.96±3.92 | 59.92±8.99 |
| StemGNN [10] | 82.36±2.23 | 377.25±3.91 | 67.90±5.94 | 42.98±2.80 | 103.15±1.87 | 63.47±4.11 | 9.57±0.80 | 51.00±2.60 | 59.60±8.81 |
| Z-GCNETs [20] | 81.22±2.15 | 356.35±3.20 | 62.81±5.75 | 43.52±2.20 | 106.22±1.27 | 65.89±4.66 | 9.37±0.50 | 46.04±1.78 | 59.21±8.73 |
| **ZFC-SHCN (ours)** | **79.33±1.70** | ****343.92±3.66** | **61.25±5.80** | **42.74±2.24** | **101.65±1.53** | **61.90±4.07** | ****9.20±0.39** | ***27.81±1.72** | **55.34±8.60** |

for GCC for $VAR_{T_1}$ and $VAR_{T_2}$ are 4.96 and 5.87, respectively; while the average EPC for $VAR_{T_1}$ and $VAR_{T_2}$ are 7.47 and 6.91, respectively. Interestingly (although it could be expected), higher GCC and lower EPC tend to be associated with higher relative gains delivered by ZFC-SHCN. Finally, in Appendix **??**, we present the sensitivity analysis for ZFC as a function of the covariance matrix in VAR models.

## 5.5 Ablation studies

To evaluate the performance of different components in our ZFC-SHCN model, we perform an expansive ablation study. The ablation study is conducted with three setups: (i) ZFC-SHCN without graph convolution in spatial dimension (W/o Graph convolution in spatial dimension), (ii) ZFC-SHCN without ZFC convolution (W/o ZFC convolution), and (iii) ZFC-SHCN without supra-Hodge convolution (W/o Supra-Hodge convolution).

The experimental results are shown in Table 4 and prove the validity of each component. As Table 4 indicates, compared to ZFC-SHCN w/o ZFC convolution, the zigzag homological feature is vital for capturing the topological structure of spatio-temporal graph and our proposed graph convolution operation on ZFC significantly improves forecasting performance. By comparing to ZFC-SHCN w/o supra-

Table 4: Ablation study on PeMSD4 and COVID-19 hospitalizations in TX.

| | Architecture | RMSE |
|---|---|---|
| PeMSD4 | **ZFC-SHCN** | ***28.58±0.15** |
| | W/o Graph convolution in spatial dimension | 28.82±0.28 |
| | W/o ZFC convolution | 28.66±0.20 |
| | W/o Supra-Hodge convolution | 28.69±0.19 |
| TX-COVID-19 | **ZFC-SHCN** | ***27.81±1.72** |
| | W/o Graph convolution in spatial dimension | 29.17±1.73 |
| | W/o ZFC convolution | 29.75±1.96 |
| | W/o Supra-Hodge convolution | 30.11±1.77 |

Hodge convolution, we illustrate the significance of higher-order structure representation learning for guiding the model to how to capture information on higher-order interactions. Also, ZFC-SHCN w/o graph convolution in spatial dimension demonstrates that the learned graph obtained from trainable weights can learn hidden information and enhance (multivariate) time-series representation learning.

## 5.6 Computational complexity

For higher-order simplices, the incidence matrices $B_1$ and $B_2$ can be calculated efficiently with complexity $\mathcal{O}(N + M)$ and $\mathcal{O}(M + Q)$ respectively, where $N$ is the number of 0-simplices (i.e., nodes), $M$ is the number of 1-simplices (i.e., edges), and $Q$ is the number of 2-simplices (i.e., filled triangles). The computational complexity of ZFC is $\mathcal{O}(\Upsilon^\delta)$ [2, 22], where $\Upsilon$ represents the number of points in time interval and $\delta \in [2, 2.373)$. The computational complexity of the overall approach is $\mathcal{O}(N^2 + \Upsilon^\delta + \Xi_k \omega F_k d_{out} + \Xi_k \omega^2 d_{out}/2 + d_{out} \sum_{\ell=t-w}^{t-1} \Xi_{k+1}^{(\ell)} + W_{GRU})$, including (i) graph convolution in spatial dimension: $\mathcal{O}(N^2)$, (ii) zigzag filtration curve: $\mathcal{O}(\Upsilon^\delta)$, (iii) supra-Hodge convolution in temporal dimension: $\mathcal{O}(\Xi_k \omega F_k d_{out} + \Xi_k \omega^2 d_{out}/2 + d_{out} \sum_{\ell=t-w}^{t-1} \Xi_{k+1}^{(\ell)})$ (where $F_k$ is the number of $k$-simplex attribute features, $\omega$ is the sliding window size, $d_{out}$ is the output dimension of the supra-Hodge convolution layer, and $\Xi_{k+1}^{(\ell)}$ is the number of $(k+1)$-simplex $\Xi_{k+1}$ at the $\ell$-th layer), and (iv) GRU: $\mathcal{O}(W_{GRU})$. We also compare our ZFC-SHCN with the most recent approach based on multipersistence-GNN [19] (i.e., TAMP-S2GCNets). We find that ZFC-SHCN yields either on-par or more competitive performance than TAMP-S2GCNets, while our proposed ZFC-SHCN significantly improves the computational efficiency (see Appendix **??** for more details). More details about running time comparison can be found in Appendix **??**.

## 6 Conclusion

We have proposed a novel framework for time-aware deep learning of time-evolving objects which takes advantages of both the higher-order interactions among the data substructures, described as simplices, and the most intrinsic time-conditioned topological information exhibited by the object, characterized via zigzag persistent homology. By leveraging the power of simplicial convolution operation and zigzag persistence for time-indexed data, ZFC-SHCN has been shown to demonstrate capabilities to yield the most competitive forecasting performance, while requiring fewer computational resources than its closest competitors. Still, computational complexity and limited theoretical results on statistical inference for zigzag persistence remain one of the major existing limitations of ZFC and, more generally, all topological methods for time dependent processes. In the future, we plan to investigate these theoretical and methodological challenges and will extend the ZFC-SHCN idea to anomaly detection in streaming time-dependent processes.

## Acknowledgments

This work was partially supported by the National Science Foundation (NSF) under awards # ECCS 2039701 and # ECCS-2039716, the Department of the Navy, Office of Naval Research (ONR) under ONR award # N00014-21-1-2530, C3.ai Digital Transformation Institute, and NASA AIST grant 21-AIST21_2-0059. Part of this material is also based upon work supported by (while serving at) the NSF. Any opinions, findings, and conclusions or recommendations expressed in this material are those of the author(s) and do not necessarily reflect the views of the NSF, ONR, C3.ai DTI, or NASA.

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
