# Appendix

## A  Proof of Proposition 3.2

Let $\eta_0 : \mathrm{DgmZZ}_{\nu_*} \mapsto \mathrm{DgmZZ}'_{\nu_*}$ be the bijective map that yields the Wasserstein-1 distance between $\mathrm{DgmZZ}_{\nu_*}$ and its perturbed copy $\mathrm{DgmZZ}'_{\nu_*}$. That is, Wasserstein-1 distance is defined as

$$\mathcal{W}_1(\mathrm{DgmZZ}_{\nu_*}, \mathrm{DgmZZ}'_{\nu_*}) = \inf_{\eta} \Big( \sum_{\mathrm{x} \in \mathrm{DgmZZ}_{\nu_*} \cup \mathcal{L}} ||\mathrm{x} - \eta(\mathrm{x})||_\infty^1 \Big),$$

where $|| \cdot ||_\infty$ is a norm in $\mathcal{L}^\infty$, i.e., $||z||_\infty = \max_i |z_i|$, $\Gamma = \{(t,t)|t \in \mathbb{R}\}$ and $\nu$ ranges over all bijections between $\mathrm{DgmZZ}_{\nu_*} \cup \Gamma$ and $\mathrm{DgmZZ}'_{\nu_*} \cup \Gamma$. Inclusion of the diagonal $\Gamma$ ensures that the set of bijections is non-empty even under different cardinalities of $\mathrm{DgmZZ}_{\nu_*}$ and $\mathrm{DgmZZ}'_{\nu_*}$.

Let $\mathrm{ZFC}_{\nu_*}(\Delta t_i^-)$ and $\mathrm{ZFC}'_{\nu_*}(\Delta t_i^-)$ be Zigzag Filtration Curves corresponding to the zigzag persistence diagrams $\mathrm{DgmZZ}_{\nu_*}$ and $\mathrm{DgmZZ}'_{\nu_*}$, respectively, and evaluated at $\Delta t_i^-$.

Then, similarly to [39], for each $i = \{1, 2, \ldots, \mathcal{N}\}$, we have

$$\left| \mathrm{ZFC}_{\nu_*}(\Delta t_i^-) - \mathrm{ZFC}'_{\nu_*}(\Delta t_i^-) \right| = \left[ \sum_{j=1}^{\mathcal{M}} \xi_i(t_{b_j}, t_{d_j}) \omega_i - \sum_{j=1}^{\mathcal{M}} \xi_i(\eta_0(t_{b_j}, t_{d_j})) \omega_i \right]$$

$$\leq \omega_i \sum_{j=1}^{M} \left[ \xi_i(t_{b_j}, t_{d_j}) - \xi_i(\eta_0(t_{b_j}, t_{d_j})) \right] \leq \omega_i L_i \sum_{j=1}^{\mathcal{M}} \left[ (t_{b_j}, t_{d_j}) - \eta_0(t_{b_j}, t_{d_j}) \right]$$

$$= \omega_i L_i \mathcal{W}_1\big(\mathrm{DgmZZ}_{\nu_*}, \mathrm{DgmZZ}'_{\nu_*}\big) < \omega_i L_i \epsilon.$$

Here in the second inequality, we take into account that $\xi_i$ is a Lipschitz continuous function with constant $L_i$. A similar result holds for $\mathrm{ZFC}_{\nu_*}(\Delta t_i^+)$ and $\mathrm{ZFC}'_{\nu_*}(\Delta t_i^+)$, $i = \{1, 2, \ldots, \mathcal{N}\}$.

Hence,

$$||\mathrm{ZFC}_{\nu_*} - \mathrm{ZFC}'_{\nu_*}||_\infty \leq C W_1\big(\mathrm{DgmZZ}_{\nu_*}, \mathrm{DgmZZ}'_{\nu_*}\big),$$

where $C = \max_{\{i=1,2,\ldots,\mathcal{N}\}} \omega_i L_i$ and $\mathrm{ZFC}_{\nu_*}$ and $\mathrm{ZFC}'_{\nu_*}$ are viewed as $(2n-1)$-dimensional vectors (i.e., ZFC is evaluated at each interval $\Delta t_i^+$ and $\Delta t_i^-$, $i = 1, 2, \ldots, \mathcal{N}$), which concludes the proof.

## B  Background on the Hodge Theory and Hodge-Laplacians

Let $C^k$ be a real-valued vector space which is endowed with basis from the oriented $k$-simplices. (By orientation of simplices, we mean selecting some (arbitrary) order for its nodes, where two orderings are said to be equivalent if they differ by an even permutation.) A linear map $\partial_k : C^k \to C^{k-1}$ is called a *boundary* operator. The adjoint of the boundary map induces the *co-boundary* operator $\partial_k^T : C^k \to C^{k+1}$. We define matrix representations of $\partial_k$ and $\partial_k^\top$ as $\boldsymbol{B}_k$ and $\boldsymbol{B}_k^\top$, respectively.

**Definition B.1.** An operator over oriented $k$-simplices $\boldsymbol{\mathcal{L}}_k : C^k \to C^k$ is called the $k$-*Hodge Laplacian*, and its matrix representation is given by

$$\boldsymbol{\mathcal{L}}_k = \boldsymbol{B}_k^\top \boldsymbol{B}_k + \boldsymbol{B}_{k+1} \boldsymbol{B}_{k+1}^\top, \tag{7}$$

where $\boldsymbol{B}_k^\top \boldsymbol{B}_k$ and $\boldsymbol{B}_{k+1} \boldsymbol{B}_{k+1}^\top$ are often referred to $\boldsymbol{\mathcal{L}}_k^{down}$ and $\boldsymbol{\mathcal{L}}_k^{up}$, respectively.

That is, the standard graph Laplacian $\boldsymbol{\mathcal{L}}_0 = \boldsymbol{B}_1 \boldsymbol{B}_1^\top \in \mathbb{R}^{N \times N}$ is a special case of the above $k$-th combinatorial Hodge Laplacian and the matrix $\boldsymbol{\mathcal{L}}_1 \in \mathbb{R}^{M \times M}$ is the Hodge 1-Laplacian.

## C  Additional Experiments and Running Time

### C.1  Datasets

Detailed data description is as follows:

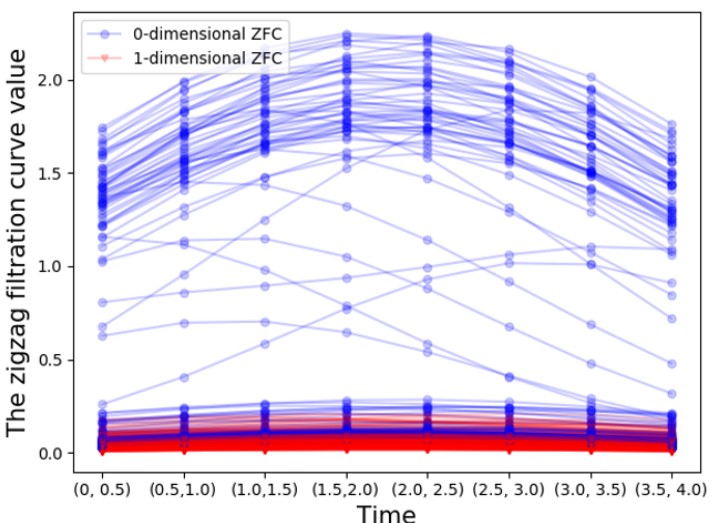

Figure 1: Illustration of 0- and 1- dimensional ZFCs for COVID-19 dataset on Texas (TX). For COVID-19 hospitalization forecasting on TX, we set the sliding window size to be 5 and thus $\mathcal{N} = 8$ over the time period $[t_1, t_5]$.

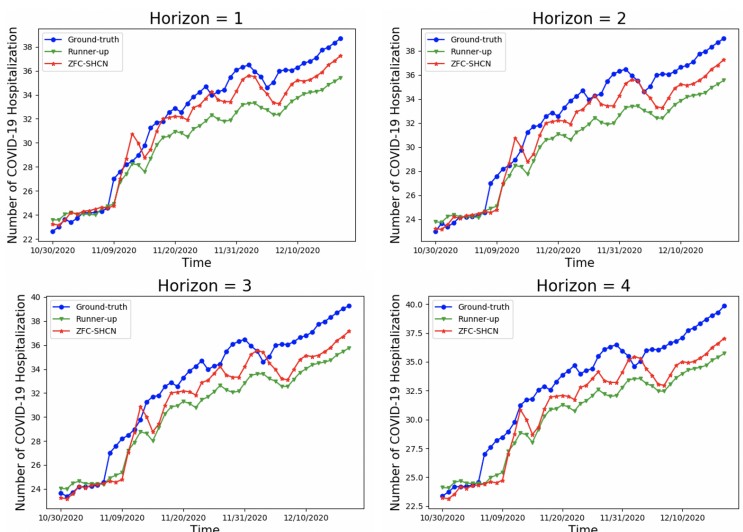

Figure 2: Comparison of forecasting performance at four different prediction horizons for COVID-19 hospitalization in TX.

1. The **COVID-19** datasets include the daily number of hospitalizations from February 1, 2020 to December 31, 2021 on a county level in three US states (i.e., CA, PA, and TX), where each node denotes a county and each edge represents a border connection between two adjacent counties. In our experiments, we split the COVID-19 datasets into training and test sets in chronological order with the split ratio $8 : 2$ for all three states.

2. The **traffic** datasets, i.e., PeMSD4 and PeMSD8 consist of the traffic flow data in California from January 1, 2018 to February 28, 2018 and from January 7, 2016 to August 31, 2016, respectively. Moreover, both PeMSD4 and PeMSD8 are aggregated to 5 minutes, which means there are 12 time points in the flow data per hour, and we split the traffic datasets with ratio $6 : 2 : 2$ into training, validation, and test sets by following the same setting of [30].

3. The **synthetic MTS** datasets include two MTS following vector autoregressions $\text{VAR}_{T_1}$: $\{\Xi_t\}_{t=1}^{T_1} = \{\xi_{1t}, \ldots, \xi_{pt}\}_{t=1}^{T_1}$ and $\text{VAR}_{T_2}$: $\{\Xi_t\}_{t=1}^{T_2} = \{\xi_{1t}, \ldots, \xi_{pt}\}_{t=1}^{T_2}$ with the graph node size $p = 20$ for two different time series lengths ($T_1 = 200$ and $T_2 = 300$). Note that

Table 5: Summary of the time series forecasting datasets. $^{\dagger}$ represents the average number of edges in simulated VAR datasets under threshold $\omega_{\text{VAR}} = 0.1$.

| Dataset | # Nodes | # Edges | Time range | # Timesteps |
|---------|---------|---------|------------|-------------|
| CA | 55 | 129 | 02/01/2020 - 12/31/2020 | 335 |
| PA | 60 | 139 | 02/01/2020 - 12/31/2020 | 335 |
| TX | 251 | 709 | 02/01/2020 - 12/31/2020 | 335 |
| PeMSD4 | 307 | 340 | 01/01/2018 - 02/28/2018 | 16,992 |
| PeMSD8 | 170 | 274 | 01/07/2016 - 08/31/2016 | 17,856 |
| VAR$_{T_1}$ | 20 | 85$^{\dagger}$ | - | 100 |
| VAR$_{T_2}$ | 20 | 73$^{\dagger}$ | - | 200 |
| Bytom | 100 | 10 | 07/27/2017 - 05/07/2018 | 285 |

for two VAR datasets we split into training and test sets with the ratio $8 : 2$. We define the $\omega$-lag vector autoregressive (VAR($\omega$)) $\boldsymbol{\Xi}_t = \{\xi_{1t}, \ldots, \xi_{pt}\}^{\top}$ as follows

$$\boldsymbol{\Xi}_t = \boldsymbol{a} + \boldsymbol{\Pi}_1 \boldsymbol{\Xi}_{t-1} + \boldsymbol{\Pi}_2 \boldsymbol{\Xi}_{t-2} + \cdots + \boldsymbol{\Pi}_w \boldsymbol{\Xi}_{t-w} + \boldsymbol{\epsilon}_t,$$

where $\boldsymbol{\Pi}_i \in \mathbb{R}^{q \times q}$ is an coefficient matrix and $\boldsymbol{\epsilon}_t \in \mathbb{R}^{q \times 1}$ denotes an unobservable mean white noise vector process with time invariant covariance matrix $\boldsymbol{\Sigma}$. In our experiments, we set $q = 20$ and $\omega = 10$.

4. The **surface air temperature** datasets include the daily surface air temperature from February 1, 2020 to December 31, 2021 on a county level in three US states (i.e., CA, PA, and TX). Besides, we use the same experimental settings as the COVID-19 datasets.

5. The **Ethereum blockchain Bytom networks** are compound of addresses of users (nodes) and daily transactions among users (edges) [24, 2]. Since original token networks have an average of 442788/1192722 nodes/edges, we compute a subgraph via a maximum weight subgraph approximation [58] using the amount of transactions as weight. The dynamic networks contain different number of networks since every token was created at different days. Hence, for each timestamp $t$ (from July 27, 2017 to May 07, 2018), given the dynamic network $\mathcal{G}_t = \{\mathcal{V}_t, \mathcal{E}_t, \tilde{W}_t\}$ and its corresponding node feature matrix $X_t \in \mathbb{R}^{N \times F}$, where $F$ represents the number of features (e.g., node degree, node betweenness, etc), we test our algorithm with both node and edge features and use the set of most active nodes, i.e., $N = 100$.

## C.2 Surface Temperature Prediction Performance

We further compare our ZFC-SHCN with the 3 strongest SOA baselines for surface temperature forecasting on CA, PA, and TX in Table 6. The results show that ZFC-SHCN achieves relative gains of 39.55%, 6.02%, and 15.59% over runner-ups (see the results with *dotted underline*) on CA, PA, and TX respectively.

Table 6: Forecasting results (MAPE (%)) on surface temperature in CA, PA, and TX.

| Model | CA | PA | TX |
|-------|-----|-----|-----|
| AGCRN | 4.21±1.05 | 7.07±1.21 | 12.97±1.39 |
| StemGNN | 4.59±1.21 | 6.61±1.10 | 11.94±1.60 |
| Z-GCNETs | 4.34±0.87 | 6.52±1.05 | 12.00±1.65 |
| **ZFC-SHCN (ours)** | **3.11±0.65 | **6.15±1.15** | *10.33±1.57** |

## C.3 Wind Speed Prediction Performance

To assess the utility of our approach for other tasks, we now also conducted spatio-temporal forecasting for graph values (i.e., on graph-level) on the wind speed dataset [28]. In this experiment, we use hourly wind speed data to predict the total wind speed from 57 stations in East Coast including

Massachusetts, Connecticut, New York, and New Hampshire. The evaluation results are summarized in Table 3. We find that our ZFC-SHCN outperforms the next best competitor, with up to 39.20% of improvement, and the results are highly statistically significant.

Table 7: Forecasting results of ZFC-SHCH and baselines on wind speed dataset.

|  | Model | RMSE |
|---|---|---|
|  | **ZFC-SHCN** | ***$1.07 \pm 0.02$** |
|  | ZPI-SHCN | $1.30 \pm 0.02$ |
| **Wind Speed** | Z-GCNETs | $1.68 \pm 0.05$ |
|  | AGCRN | $1.52 \pm 0.03$ |
|  | StemGNN | $1.76 \pm 0.03$ |

## C.4 Additional Ablation Studies

We conduct the additional experiments by (i) replacing ZFC with zigzag persistence image (ZPI), i.e., ZPI-SHCN and (ii) replacing Supra-Hodge convolution with GCN (which is temporal graph convolution in temporal domain), i.e., ZFC-GCN, on both PeMDS4 and TX-COVID-19 (in Texas state) datasets. First, Table 8 demonstrates that for both datasets, the ZFC representation (i.e., ZFC-SHCN) significantly outperforms zigzag persistence image (i.e., ZPI-SHCN). These findings might be explained by the fact that, especially for small- and medium-size datasets, ZFC tends to better capture the topological signal than ZPI which, being a 2-d representation, might be more noisy. Second, Table 8 indicates that compared to ZFC-GCN, ZFC-SHCN always achieves better forecasting performance. We attribute these findings to the fact that supra-Hodge convolution operation integrates knowledge on important interactions among higher-order structures (i.e., edges, filled triangles - beyond the node-space) into graph learning. For TX-COVID-19 dataset, the relative gain of ZFC-SHCN over the runner-up (i.e., ZFC-GCN) in RMSE is 3.13%; for PeMSD4 dataset, the relative gain of ZFC-SHCN over the runner-up (i.e., ZFC-GCN) in RMSE is 1.11%.

Table 8: Additional ablation study of ZFC convolution and Supra-Hodge convolution on PeMSD4 and COVID-19 hospitalizations in TX.

|  | Architecture | RMSE |
|---|---|---|
|  | **ZFC-SHCN** | ***$28.58 \pm 0.15$** |
| **PeMSD4** | ZPI-SHCN | $29.00 \pm 0.17$ |
|  | ZFC-GCN | $28.90 \pm 0.26$ |
|  | **ZFC-SHCN** | **$27.81 \pm 1.72$** |
| **TX-COVID-19** | ZPI-SHCN | $29.26 \pm 1.86$ |
|  | ZFC-GCN | $28.71 \pm 1.90$ |

Regarding the choice of simplicial complexes, we have also conducted additional experiments between (i) ZFC-SH$^1$CN based on Supra-Hodge 1-Laplacian (i.e., utilizing information of 0-simplices, 1-simplices, and 2-simplices) and (ii) ZFC-SH$^2$CN based on Supra-Hodge 2-Laplacian (i.e., utilizing information of 1-simplices, 2-simplices, and 3-simplices) on PeMSD4 and COVID-19 in TX dataset. The evaluation results are summarized in Table 9. Experimental results show that ZFC-SH$^1$CN consistently outperforms ZFC-SH$^2$CN. We tend to attribute such findings to the fact that we observe a very low number of 3-simplices. For instance, in PeMSD4 at each timestamp we observe around 29 2-simplices (filled triangles) but we observe only around 2 3-simplices (tetrahedron). Furthermore, it is worth to mention that: for PeMSD4 dataset, ZFC-SH$^2$CN is still very competitive compared to other baselines (i.e., slightly worse than the runner-up, i.e., Z-GCNETs); for COVID-19 in TX, the performance of ZFC-SH$^2$CN is much better the runner-up. Nevertheless, we believe that higher-order simplices, if observed in the targeted network, has a potential to boost the model performance.

Table 9: Additional ablation study of simplices on PeMSD4 and COVID-19 hospitalizations in TX.

|  | Architecture | RMSE |
|---|---|---|
| **PeMSD4** | **ZFC-SH$^1$CN** | ***$\mathbf{28.58 \pm 0.15}$ |
|  | ZFC-SH$^2$CN | $29.15 \pm 0.30$ |
| **TX-COVID-19** | **ZFC-SH$^1$CN** | **$\mathbf{27.81 \pm 1.72}$ |
|  | ZFC-SH$^2$CN | $29.39 \pm 1.93$ |

## C.5 Comparison to Multiparameter Persistence

The development of multiparameter persistence (MP) has recently witnessed a rapid progress (such as [14, 20]) due to in many applications, particularly, involving spatio-temporal processes, the data exhibit richer structures which cannot be well encoded with a single parameter persistence. Although there exist some studies in the literature that have shown promising performances for spatial temporal prediction tasks [20], the computation of MP is labor-intensive and very time-consuming. To better demonstrate the power of our proposed ZFC-SHCN, here we compare our method with one representative MP-based GNNs, i.e., TAMP-S2GCNets [20] from both prediction performance and running time perspectives. We conduct comparison experiments on PeMSD4, CA, and Bytom, and we find that ZFC-SHCN yields highly competitive performance against TAMP-S2GCNets (see Figures 3a and 4a); that is, for **PeMSD4**, ZFC-SHCN: {MAE: 17.58±0.17, RMSE: 28.58±0.15, MAPE (%): 11.68±0.07} vs. TAMP-S2GCNets: {MAE: 17.58±0.20 RMSE: 28.56±0.28, MAPE (%) 11.01}; for **CA**: ZFC-SHCN: {MAE: 79.33±1.70, RMSE: 343.92±3.66, MAPE (%): 61.25±5.80} vs. TAMP-S2GCNets: {MAE: 82.75±1.93, RMSE: 371.60±2.68, MAPE (%): 62.43±5.61}; for **Bytom**, ZFC-SHCN: {MAPE: 29.51±0.77} vs. TAMP-S2GCNets: {MAPE: 29.26±1.06}). Furthermore, Figures 3b and 4b present visual comparison of running time of ZFC-SHCN and TAMP-S2GCNets (per epoch in seconds). We find that training time consumed by ZFC-SHCN is substantially lower than that of TAMP-S2GCNets on all datasets. These findings suggest that ZFC-SHCN may be the preferred forecasting choice, in terms of the balance between accuracy and computational efficiency.

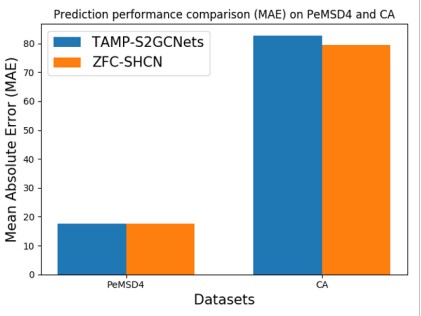
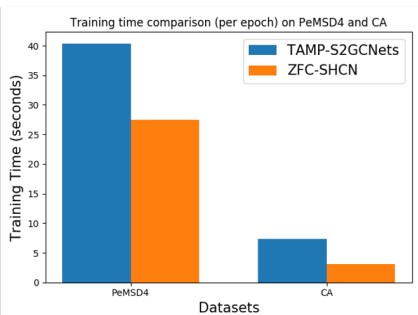

(a) Prediction performance comparison.

(b) Training time comparison.

Figure 3: Prediction performance (RMSE) and training time comparisons on PeMSD4 and CA.

## C.6 Overall Comparison with Runner-ups

We present comparison of computational time of our ZFC-SHCN with the 3 strongest SOA baselines on PeMSD4, CA, and Bytom in Table 10. The results show the superior computational efficiency of ZFC-SHCN. Note, that while AGCRN is faster than ZFC-SHCN, ZFC-SHCN significantly outperforms AGCRN in terms of forecasting accuracy, e.g., with a relative gain of up to 48.81% on MAPE for CA. Furthermore, from Tables 11 and 12, we find both the running times on zigzag-based topological feature generation and per-epoch training time of ZFC-SHCN are less than the runner-up model (i.e., Z-GCNETs).

Furthermore, we have conducted comparison experiments between our ZFC-SHCN and two baselines, i.e., fractional-order dynamical model [31] and Padé Exp [32] on COVID-19 (TX) dataset. Table 13 shows that ZFC-SHCN outperforms both fractional-order dynamical model and Padé Exp.

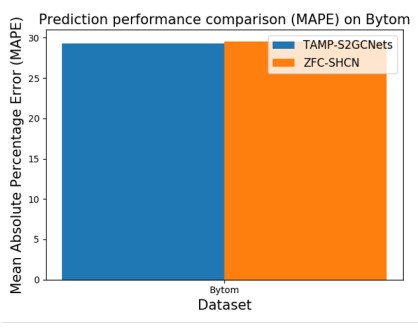

(a) Prediction performance comparison.

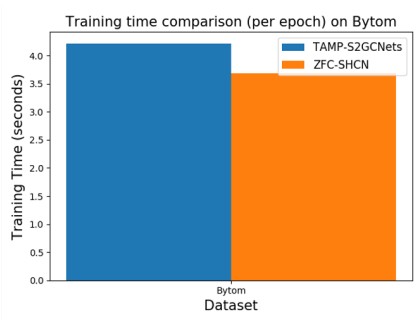

(b) Training time comparison.

Figure 4: Prediction performance (MAPE) and training time comparisons on Bytom.

Table 10: Comparisons of prediction performance and computation time (in ( )) on PeMSD4, CA, and Bytom datasets.

| Dataset | RMSE/MAPE (Average Training Time per Epoch (sec)) | | |
| | PeMSD4 (*RMSE*) | CA (*RMSE*) | Bytom (*MAPE*) |
|---|---|---|---|
| AGCRN | 29.17±0.09 (25.74 s) | 448.27±2.78 (3.01 s) | 34.46±1.37 (2.47 s) |
| StemGNN | 31.83±0.11 (33.86 s) | 377.25±3.91 (6.63 s) | 34.50±1.53 (3.05 s) |
| Z-GCNETs | 29.08±0.19 (30.12 s) | 356.35±3.20 (6.56 s) | 31.04±0.78 (2.83 s) |
| **ZFC-SHCN (ours)** | ***28.58±0.15** (27.53 s) | ****343.92±3.66** (3.15 s) | ***29.51±0.77** (2.82 s) |

## C.7  Sensitivity analysis of the covariance matrix

To further investigate the effect of covariance matrix in ZFC's representational power, we perform the sensitivity analysis of covariance matrix $\Sigma$ of ZFC, which specifies the patterns of variability as well as the shape of the multivariate Gaussian distribution. We show that the variation of RMSE for COVID-19 hospitalization prediction task with $\Sigma$ (note that, we assume $\Sigma$ is the same for every persistent point $(t_{b_j}, t_{d_j})$). Table 14 shows variations of RMSE with different diagonal and off-diagonal elements in $\Sigma$. As demonstrated in Table 14, (i) RMSE increases with increasing the off-diagonal element (i.e., the correlation coefficient) when fixing diagonal element; (ii) RMSE first decreases and then increases via increasing diagonal element when setting the correlation coefficient to be 0 (i.e., there is no linear relationship between the $t_{b_j}$ and $t_{d_j}$). In addition, we find evidence of a non-monotonic relationship between prediction performance (in RMSE) and diagonal element of $\Sigma$.

# D  Broader Impact

The key **positive** impact of this project constitutes in the simultaneous enhancement of both predictive performance and computational efficiency in forecasting tasks associated with complex multivariate time series, including but not limited to spatio-temporal processes. Such tasks are widely met in a diverse range of applications, from finance to atmospheric sciences to biosurveillance. Furthermore, topological methods and graph representations of multivariate time series and spatio-temporal processes, in conjunction with forecasting, remains yet nascent in both machine learning and statistics, and has a high potential, especially for analysis of multivariate time series with a complex dependence structure, such as disease surveillance, wildfire monitoring, and cryptomarket analytics. However, the current lack of proper statistical inferential methods for topological characteristics of multivariate

Table 11: Computational costs for generation of zigzag persistence images (ZPI) and a single training epoch of Z-GCNETs.

| Dataset | Average Time Taken (sec) | |
| | ZPI | Z-GCNETs (epoch) |
|---|---|---|
| CA | 0.23 s | 6.56 s |
| PeMSD4 | 0.86 s | 30.12 s |

Table 12: Computational costs for generation of zigzag filtration curve (ZFC) and a single training epoch of ZFC-SHCN.

| Dataset | Average Time Taken (sec) | |
| | ZFC | ZFC-SHCN (epoch) |
| --- | --- | --- |
| CA | 0.10 s | 3.15 s |
| PeMSD4 | 0.52 s | 27.53 s |

Table 13: Comparison of RMSE of ZFC-SHCN and fractional-order dynamical model on COVID-19 in TX.

| | Architecture | RMSE |
| --- | --- | --- |
| **TX-COVID-19** | **ZFC-SHCN** | **27.81±1.72** |
| | Fractional-order dynamical model | 28.60±1.88 |
| | Padé Exp | 28.21±1.69 |

Table 14: Sensitivity analysis of the covariance matrix $\Sigma$ of ZFC on COVID-19 hospitalizations in CA and TX.

| Covariance Matrix | CA | TX |
| --- | --- | --- |
| $\Sigma = \begin{bmatrix} 1 & 0 \\ 0 & 1 \end{bmatrix}$ | 346.66±2.78 | 30.16±3.61 |
| $\Sigma = \begin{bmatrix} 1 & 0.1 \\ 0.1 & 1 \end{bmatrix}$ | 349.88±2.67 | 30.22±3.02 |
| $\Sigma = \begin{bmatrix} 1 & 0.5 \\ 0.5 & 1 \end{bmatrix}$ | 350.94±1.95 | 32.50±3.47 |
| $\Sigma = \begin{bmatrix} 1 & 0.8 \\ 0.8 & 1 \end{bmatrix}$ | 352.76±2.95 | 32.83±2.83 |
| $\Sigma = \begin{bmatrix} 10 & 0 \\ 0 & 10 \end{bmatrix}$ | 347.88±2.89 | 31.32±2.55 |
| $\Sigma = \begin{bmatrix} 5 & 0 \\ 0 & 5 \end{bmatrix}$ | **343.92±3.66** | **27.81±1.72** |

time series makes it questionable how **robust** these methods and the conclusions derived from them are, especially under adverse scenarios. This **negative** impact of the developed predictive analytics shall be necessarily accounted in the proper risk management, particularly, when targeted attacks are expected, and shall be further investigated with rigours inferential tools of mathematical statistics.