# OpenReview forum: "Time-Conditioned Dances with Simplicial Complexes: Zigzag Filtration Curve based Supra-Hodge Convolution Networks for Time-series Forecasting"
_NeurIPS.cc/2022/Conference — NeurIPS 2022 Accept_

### Official Review · Reviewer_vJan · 2022-07-11

**Rating:** 5
**Confidence:** 3
**Soundness:** 2 fair
**Presentation:** 2 fair
**Contribution:** 3 good

**Summary:**

The authors propose the Zigzag Filtration Curve based Supra-Hodge Convolution Networks (ZFC-SHCN) which is able to learn time-aware persistent topological features and simplicial geometry of graphs. The theoretical stability of the zigzag filtration curve is studied in the paper. Experiment results show that ZFC-SHCN achieves the best performance of forecasting with less computational requirements.


**Questions:**

Please see weakness.

**Limitations:**

 Limitations and potential negative societal impact have been addressed.

**Strengths And Weaknesses:**

Strengths:
- The proposed novel framework connects the dynamical behavior of complex systems and persistence homology analysis.
- The authors claim that this paper is the first work that brings the concepts of simplicial convolution to time-aware learning.
- The experiments on varying datasets including covid case and traffic flow show the superiority of the proposed method in terms of the prediction tasks. ZFC-SHCN is also faster compared with runner-ups.

Weakness:
- I recommend adding some references in the introduction section to support the statements.
- I suggest the authors to check these related paper of time series prediction:
https://ieeexplore.ieee.org/abstract/document/8430866
https://openreview.net/forum?id=d2TT6gK9qZn
- The writing quality of this paper needs to be improved, please see below some questions that need to be clarified in the paper main body.
- The contribution is clear as stated in the introduction section. However, the connection among/inside section 3 and 4 (and the subsections) is not very clear.
- Are the tasks to predict the further covid cases, traffic flow etc. as both time series and node attribute of graph?
- How are the graphs or the simplicial complexes constructed on each dataset? In the appendix it says the graph represents the border connection, but it is unclear about other datasets.
- How is the simplex constructed in the experiment section? Does it only consider nodes and edges, or also involve actual higher-order information like a 2-simplex, 3-simplex or more? If not, where does the higher-order information appear in the graph structure?
- Is the embedding dimension mentioned in line 281 the same as the dimension of feature of each node (d_e) or something else? Is it predefined or computed following any rules? The number here looks rather small, any reason for such a choice?

---

> ### Author Response · Authors · 2022-07-31
> **Response to Technical Questions (3/3)**
>
> Q6: ``How is the simplex constructed in the experiment section? Does it only consider nodes and edges, or also involve actual higher-order information like a 2-simplex, 3-simplex or more? If not, where does the higher-order information appear in the graph structure?''
>
> $\bf{A}$: In our experiments, after the construction of target spatio-temporal graphs, we extract 0-simplices, 1-simplices, and 2-simplices for each spatio graph $\mathcal{G}_t$ at timestamp $t$. In order to integrate these simplicial complexes into our model, we firstly generate Hodge-$k$ Laplacian via $\boldsymbol{L}\_k =\boldsymbol{B}\_k^{\top}\boldsymbol{B}\_k+\boldsymbol{B}\_{k+1}\boldsymbol{B}\_{k+1}^{\top}$ (where $\boldsymbol{B}\_k$ and $\boldsymbol{B}\_{k+1}$ are the $(k-1)$-simplex-to-$k$-simplex and $k$-simplex-to-$(k+1)$-simplex incidence matrices respectively); then, according to Equation 3 (see Section 4.3), we can obtain supra-Hodge $k$-Laplacian (here $k=1$). Finally, we feed the Supra-Hodge $k$-Laplacian into our diffusion supra-Hodge convolution operation to to capture time-conditioned higher-order interactions on $\mathcal{G}_t$ and to describe the dynamic diffusion of information over simplices along the temporal dimension. In our experiments, we consider 0-simplices (nodes), 1-simplices (edges), and 2-simplices (filled triangles).
> Thank you for suggesting to explore the potential utility of 3-simplices! Based on 1-simplices (edge), 2-simplices (filled triangles), and 3-simplex (tetrahedron) and the definition of Hodge-$k$ Laplacian, we can obtain Hodge 2-Laplacian and the corresponding Supra-Hodge $2$-Laplacian (via Equation~3 in Section 4.3). After that, we conduct comparison experiments between (i) ZFC-SH$^1$CN based on supra-Hodge $1$-Laplacian (i.e., utilizing information of 0-simplices, 1-simplices, and 2-simplices) and (ii) ZFC-SH$^2$CN based on supra-Hodge $1$-Laplacian (i.e., utilizing information of 1-simplices, 2-simplices, and 3-simplices). Tables 13 and 14 present the performances of ZFC-SH$^1$CN and ZFC-SH$^2$CN on PeMSD4 and COVID-19 (TX) datasets. We find that ZFC-SH$^1$CN consistently outperforms ZFC-SH$^2$CN.
> We tend to attribute such findings to the fact that we observe a very low number of 3-simplices. For instance, in PeMSD4 at each timestamp we observe around $\bf{29}$ 2-simplices (filled triangles) but we observe only around $\bf{2}$ 3-simplices (tetrahedron). Nevertheless, we believe that higher order simplices, if observed in the targeted network, has a potential to boost the model performance.
>
> $\bf{Table}$ $\bf{13}$: Comparison between ZFC-SH$^1$CN vs ZFC-SH$^2$CN on PeMSD4.
>
> +++++++++++++++++++++++++++++++++
>
> Dataset | Architecture | RMSE
>
> +++++++++++++++++++++++++++++++++
>
> PeMSD4 | ZFC-SH$^1$CN | $^{***}$28.58$\pm$0.15
>
> +++++++++++++++++++++++++++++++++
>
> PeMSD4 | ZFC-SH$^2$CN | 29.15$\pm$0.30
>
> +++++++++++++++++++++++++++++++++
>
> $\bf{Table}$ $\bf{14}$: Comparison between ZFC-SH$^1$CN vs ZFC-SH$^2$CN on COVID-19 in TX.
>
> +++++++++++++++++++++++++++++++++
>
> Dataset $\hspace{2em}$| Architecture | RMSE
>
> +++++++++++++++++++++++++++++++++
>
> TX: COVID-19 | ZFC-SH$^1$CN | $^{**}$27.81$\pm$1.72
>
> +++++++++++++++++++++++++++++++++
>
> TX: COVID-19 | ZFC-SH$^2$CN | 29.39$\pm$1.93
>
> +++++++++++++++++++++++++++++++++
>
> Q7: ``Is the embedding dimension mentioned in line 281 the same as the dimension of feature of each node (d_e) or something else? Is it predefined or computed following any rules? The number here looks rather small, any reason for such a choice?''
>
> $\bf{A}$: The embedding dimension $d_c \in \{1, 2, 3, 5, 10\}$ mentioned in line 281 is the dimension of node embedding dictionary $\boldsymbol{W}^{\phi}$ which helps construct a self-adaptive adjacency matrix $\boldsymbol{S}$ (defined in Section 4.1). The dimension of feature of each node $d_e$ depends on the target dataset. For traffic datasets (PeMSD4 and PeMSD8), the dimension of feature of each node $d_e$ is 3, i.e., speed, occupancy and flow rate; for COVID-19 hospitalizations datasets (CA, PA, and TX), the dimension of feature of each node $d_e$ is 1, i.e., COVID-19 cases; for surface air temperature datasets (CA, PA, and TX), the dimension of feature of each node $d_e$ is 1, i.e., surface air temperature (see results in Appendix C.2); for synthetic multivariate time-series (MTS) datasets, the dimension of feature of each node $d_e$ is 1, i.e., simulated value based on vector autoregressive (VAR); for Blockchain (i.e., Bytom) dataset, the dimension of feature of each node $d_e$ is 1, i.e., node degree. The range embedding dimension $d_c$ is initially set within {1,2,3,4, 5, 6, 7, 8, 9, 10, 20, 30, 40, 50, 60, 70, 80, 90, 100}. Throughout these experiments, we find that large node embedding dimensions will always lead to weaker performance. Therefore, we consider small node embedding dimensions in our study.

---

> > ### Comment · Reviewer_vJan · 2022-08-09
> > **Thanks for the response from the authors**
> >
> > Thanks for the response, it answers most of my questions. I would like to revise the rating score.

---

> > > ### Author Response · Authors · 2022-08-09
> > > **Thanks very much!**
> > >
> > > Thanks very much for the very constructive and valuable feedback inspiring us to think on new directions and for raising the score!

---

> ### Author Response · Authors · 2022-07-31
> **Response to Technical Questions (2/3)**
>
> Q4: ``Are the tasks to predict the further covid cases, traffic flow etc. as both time series and node attribute of graph?''
>
> $\bf{A}$: Yes, in this paper, we target on multi-step spatio-temporal (traffic and COVID-19) forecasting problem. Given observed values of past $\omega$ time periods, i.e., $\boldsymbol{X}^{\omega}$ = {$\boldsymbol{X}\_{t-\omega+1}, \dots, \boldsymbol{X}\_{t}$} $\in \mathbb{R}^{\omega \times N \times F}$ ($F$ is the node feature dimensionality), our task of multi-step spatio-temporal forecasting aims to predict the node values/attributes in multivariate spatio-temporal graph $\mathcal{G} = (\mathcal{V}, \mathcal{E}, \boldsymbol{X})$ for next $h$ timestamps, i.e., {$\boldsymbol{X}\_{t+1}, \dots, \boldsymbol{X}\_{t+h}$}.
> During the rebuttal stage, we have also conducted spatio-temporal forecasting for graph values (i.e., on graph-level) on wind energy dataset [5]. In this experiment, we use hourly wind speed data to predict the total wind speed from 57 stations in East Coast including Massachusetts, Connecticut, New York, and New Hampshire. The evaluation results are summarized in Table 12. We find that our ZFC-SHCN outperforms the next best competitor, with up to 39.20\% of improvement, and the results are highly statistically significant. Therefore, we believe that the highly competitive performances of ZFC-SHCN for both node values and graph value forecasting in spatio-temporal networks, makes the expression sufficiently convincing for a broad range of forecasting tasks. In the future, we will also explore the utility of ZFC-SHCN for inductive link prediction.
> Note that, we also perform a one-sided two-sample $t$-test between the best result and the best performance achieved by the runner-up, where $*, **, ***$ denote $p$-value $< 0.1, 0.05, 0.01$ (i.e., denote significant, statistically significant, highly statistically significant results, respectively.
>
> $\bf{Table}$ $\bf{12}$: Forecasting results of ZFC-SHCH and baselines on wind energy dataset.
>
> +++++++++++++++++++++++++++++++++++++++++++
>
> Dataset $\hspace{1.8em}$ | $\hspace{0.5em}$ Model $\hspace{0.5em}$ | RMSE
>
> +++++++++++++++++++++++++++++++++++++++++++
>
> Wind Energy | ZFC-SHCN | $^{***}$1.07$\pm$0.02
>
> +++++++++++++++++++++++++++++++++++++++++++
>
> Wind Energy | ZPI-SHCN | 1.30$\pm$0.02
>
> +++++++++++++++++++++++++++++++++++++++++++
>
> Wind Energy | Z-GCNETs | 1.68$\pm$0.05
>
> +++++++++++++++++++++++++++++++++++++++++++
>
> Wind Energy | $\hspace{0.1em}$ AGCRN $\hspace{0.5em}$ | 1.52$\pm$0.03
>
> +++++++++++++++++++++++++++++++++++++++++++
>
> Wind Energy | StemGNN| 1.76$\pm$0.03
>
> +++++++++++++++++++++++++++++++++++++++++++
>
> [5] Ghaderi, Amir, Borhan M. Sanandaji, and Faezeh Ghaderi. "Deep forecast: Deep learning-based spatio-temporal forecasting." ICML 2017 Time Series Workshop.
>
> Q5: ``How are the graphs or the simplicial complexes constructed on each dataset? In the appendix it says the graph represents the border connection, but it is unclear about other datasets.''
>
> $\bf{A}$: For COVID-19 datasets, each county represents a node, such that $\mathcal{E} \subseteq \mathcal{V} \times \mathcal{V}$ is the edge set, whilst edges represent border connection between counties followed by county adjacency file (i.e., official County Adjacency File Record Layout- Federal Information Processing System (FIPS) Codes). More specifically, we assume that $e_{uv} = 1$ for each edge between nodes $u$ and $v$. For traffic datasets, we use the datasets provided from [6]; in the traffic network, the node is represented by the loop detector and the edge is a freeway segment between two nearest nodes. In our experiments, we consider three types of simplicial complexes, i.e., 0-simplices (nodes), 1-simplices (edges), and 2-simplices (filled triangles). For 0-simplices and 1-simplices, we can easily extract from the graph; for 2-simplices, we compute and list the faces in the target graph (see pseudo-code in Remark F.2 in Appendix - Rebuttal Revision) and we also have provided the corresponding code in dropbox link (e.g., see ``get_faces’’ function in ‘/ZFC-SHCN code and data/CA/incidence_matrix.py’).
>
> [6] Guo, Shengnan, et al. "Attention based spatial-temporal graph convolutional networks for traffic flow forecasting." AAAI 2019.

---

> ### Author Response · Authors · 2022-07-31
> **Response to Technical Questions (1/3)**
>
> Thank you very much for the detailed and constructive suggestions and for recognizing the proposed novel framework connecting the dynamical behavior of complex systems and persistence homology analysis! We also appreciate the suggested additional benchmarks that we added into our study and which further emphasized competitive predictive capabilities of ZFC-SHCN. We have addressed your questions below and also added the respective updates into the paper (see Section 1 (Q1) (main paper), Section 2 and Remark E.5 (Q2) (main paper and Appendix), Sections 3 and 4 (Q3) (main paper), Remark E.2 (Q4) in Appendix, Remark F.2 (Q5) in Appendix, Remark F.3 (Q6) in Appendix, Remark F.4 (Q7) in Appendix). We would be grateful for any additional suggestions on the experimental or methodological validation that could further convince you in the ZFC-SHCN capabilities and could boost your assessment of ZFC-SHCN.
>
> Q1: ``Adding some references in the introduction section to support the statements.''
>
> $\bf{A}$: Thank you very much for this suggestion. In the rebuttal revision, we have added more references in the introduction section to better support our statements (see Rebuttal Revision).
>
> Q2: ``I suggest the authors to check these related paper of time series prediction: https://ieeexplore.ieee.org/abstract/document/8430866 https://openreview.net/forum?id=d2TT6gK9qZn''
>
> $\bf{A}$: Thank you very much for suggesting these two relevant papers. We have checked the results on fractional-order dynamical model [3] and Padé Exp [4] and added them into the section on the related work. The authors of paper [4] do not release their source code, and we have asked for code via email but have got no reply yet. Meanwhile, we have conducted comparison experiments between our ZFC-SHCN and fractional-order dynamical model [3] on COVID-19 (TX) dataset. Table 11 shows that ZFC-SHCN outperforms fractional-order dynamical model [3]. Once we get the code of Padé Exp [4] from authors, we will add its result into our final version. Moreover, we have cited these two papers you mentioned in the rebuttal revision (please see Section 2 paragraph 1 in Rebuttal Revision).
>
> $\bf{Table}$ $\bf{11}$: Comparison of RMSE of ZFC-SHCN and fractional-order dynamical model on COVID-19 in TX.
>
> +++++++++++++++++++++++++++++++++++++++++++++++++++++
>
> Dataset $\hspace{2.3em}$| Model $\hspace{10.5em}$| RMSE
>
> +++++++++++++++++++++++++++++++++++++++++++++++++++++
>
> TX: COVID-19 | ZFC-SHCN $\hspace{9.3em}$| 27.81$\pm$1.72
>
> +++++++++++++++++++++++++++++++++++++++++++++++++++++
>
> TX: COVID-19 | Fractional-order dynamical model | 28.60$\pm$1.88
>
>  +++++++++++++++++++++++++++++++++++++++++++++++++++++
>
> [3] Gupta, Gaurav, Sergio Pequito, and Paul Bogdan. "Dealing with unknown unknowns: Identification and selection of minimal sensing for fractional dynamics with unknown inputs." IEEE ACC 2018.
>
> [4] Gupta, Gaurav, et al. "Non-linear operator approximations for initial value problems." ICLR 2022.
>
> Q3: ``The contribution is clear as stated in the introduction section. However, the connection among/inside section 3 and 4 (and the subsections) is not very clear.''
>
> $\bf{A}$: Thank you very much for this valuable suggestion to improve the paper cohesion. We have added some connections among/inside sections 3 and 4 (see Rebuttal Revision).

---

> ### Author Response · Authors · 2022-08-02
> **Response to Technical Question Regarding Padé Exp Model**
>
> Q2: ``I suggest the authors to check these related paper of time series prediction: https://ieeexplore.ieee.org/abstract/document/8430866 https://openreview.net/forum?id=d2TT6gK9qZn''.
>
> $\bf{A}$: Based on the code of Padé Exp provided by the authors of [4], we conduct an experiment on comparing our ZFC-SHCN with Padé Exp on COVID-19 (TX) dataset. As shown in Table 11, our ZFC-SHCN model outperforms both newly suggested baselines Padé Exp [4]. We have updated the comparison results in rebuttal revision (see Table 12 in Remark E.5 in Appendix). We will add the full comparative analysis on all networks into the final version.
>
> $\bf{Table}$ $\bf{11}$: Comparison of RMSE of ZFC-SHCN, fractional-order dynamical model, and Padé Exp on COVID-19 in TX.
>
> +++++++++++++++++++++++++++++++++++++++++++++++++++++
>
> Dataset $\hspace{2em}$ | Model $\hspace{10.5em}$ | RMSE
>
> +++++++++++++++++++++++++++++++++++++++++++++++++++++
>
> TX: COVID-19 | ZFC-SHCN $\hspace{9em}$ | 27.81$\pm$1.72
>
> +++++++++++++++++++++++++++++++++++++++++++++++++++++
>
> TX: COVID-19 | Fractional-order dynamical model | 28.60$\pm$1.88
>
> +++++++++++++++++++++++++++++++++++++++++++++++++++++
>
> TX: COVID-19 | Padé Exp $\hspace{9.3em}$ | 28.21$\pm$1.69
>
> +++++++++++++++++++++++++++++++++++++++++++++++++++++
>
> [3] Gupta, Gaurav, Sergio Pequito, and Paul Bogdan. "Dealing with unknown unknowns: Identification and selection of minimal sensing for fractional dynamics with unknown inputs." IEEE ACC 2018.
>
> [4] Gupta, Gaurav, et al. "Non-linear operator approximations for initial value problems." ICLR 2022.

---

> ### Author Response · Authors · 2022-08-07
> **Looking forward to hearing back from you!**
>
> Dear Reviewer vJan,
>
> We wonder whether we can provide any additional information to clarify your questions/concerns. Thank you very much again for your constructive comments and we are looking forward to your feedback!
>
> Paper8271 Authors

---

### Official Review · Reviewer_isoH · 2022-07-11

**Rating:** 6
**Confidence:** 3
**Soundness:** 3 good
**Presentation:** 3 good
**Contribution:** 3 good

**Summary:**

In this paper, the authors propose a method for predicting time series of graphs with the addition of the concept of TDA. There is no discussion of graph construction methods here, but rather the general setting of the existence of time-varying graphs. The proposed method is basically a conventional graph convolution -recurrent neural network with an additional mechanism to consider topological information. The zigzag filtration curve and supra-Hodge convolution are used as topological information. The authors also prove the stability of ZFC and demonstrate its effectiveness in comparison to conventional methods in their experiments.

**Questions:**

1. The proposed method introduces zigzag filtration features and supra-Hodge convolution features in addition to the conventional graph convolution features. Although the overall framework has been evaluated, it is not clear which features have what effect on each of them. Since it is not possible to determine whether all of the elements are necessary, I wonder if a comparison with a system that excludes any of the information would make the observation possible. Are such comparisons being made?

2. Features using Zigzag filtration Creation is a natural concept, but I am sure there are other possibilities. Are there any comparisons with other methods? Of course, since the effectiveness of this system as a whole has been demonstrated, it is only a proposal as a first step, and if improvements using other TDA methods is a future issue, that is fine.

3. The experimental results, especially in Table 3, show that the proposed method seems to create different models for different evaluation indicators for a single problem. The same model should be used to compare different indicators in the evaluation. If you want to show that each indicator can be set appropriately, you should compare them separately.


**Limitations:**

The authors have clearly defined the application, clearly described the performance, and adequately addressed the limitation of their work.


**Strengths And Weaknesses:**

Strength
-  The authors provide a framework for time series prediction of graphs that includes topological information that has not been previously taken into account.
-  Introduced zigzag filtration as topological information
- The authors have organized the mathematical background of the algorithm before introducing the algorithm.
- The authors have tested the effectiveness of this method compared to many conventional methods from multiple perspectives on multiple data sets

Weakness
- Although the authors have compared the proposed method with conventional methods that do not include topology information, they have not examined whether the proposed method is appropriate as a method that takes topology information into account.
- There are doubts about some of the experimental setups.

Overall, the prediction of graph time series is a very important issue, including from a practical viewpoint, and it is worthwhile to have provided an effective framework for this problem with the addition of TDA-based information. Since I have some doubts, I would like to make a final decision based on the answers to the following questions.

---

> ### Author Response · Authors · 2022-07-31
> **Response to Technical Questions (3/3)**
>
> Q3: ``Features using Zigzag filtration Creation is a natural concept, but I am sure there are other possibilities. Are there any comparisons with other methods? Of course, since the effectiveness of this system as a whole has been demonstrated, it is only a proposal as a first step, and if improvements using other TDA methods is a future issue, that is fine.''
>
> $\bf{A}$: Thank you very much for this thought-provoking suggestion! We have now conducted additional experiments by replacing zigzag filtration across the time dimension with other TDA methods, i.e., (i) the single sublevel filtration based on a node degree (DSF) and (ii) multiparameter persistence (MP) ([2] - combination of two filtering function, i.e., the sublevel filtrations based on a node degree and the sublevel filtrations based on a node betweenness). For fair comparison, we just replace the topological summary of zigzag filtration with other filtration/TDA  functions and keep the supra-Hodge convolution part within the model architecture. To feed the topological summary of DSF into our architecture, we convert its persistence diagram into a fixed-length vector, i.e., persistence image, and integrate the corresponding persistence image into our architecture (called DSF-SHCN). For MP, note that it is in grid representation, we directly integrate it into our architecture (called MP-SHCN). As shown in Tables 9 and 10, we find that the model with zigzag filtration (ZFC-SHCN) outperforms both DSF-SHCN and MP-SHCN. The improvement gain of ZFC-SHCN over the baselines ranges from 1.10\% to 4.15\% on PeMSD4 and COVID-19 (in TX) datasets. Furthermore, your suggestion on alternative possibilities brought us to the idea of developing zigzag landscape and zigzag silhouette, which will do in the further extension of this project. Thank you for bringing up this direction!
>
> $\bf{Table}$ $\bf{9}$: Comparison of RMSE of ZFC-SHCN to its two variants (i.e., with different topological summaries) on PeMSD4.
> +++++++++++++++++++++++++++++++++++
>
> Dataset | Architecture $\hspace{0.001em}$| RMSE
>
> +++++++++++++++++++++++++++++++++++
>
> PeMSD4 | ZFC-SHCN $\hspace{0.02em}$ | $^{***}$28.58$\pm$0.15
>
> +++++++++++++++++++++++++++++++++++
>
> PeMSD4 | DSF-SHCN $\hspace{0.02em}$  | 29.82$\pm$0.22
>
> +++++++++++++++++++++++++++++++++++
>
> PeMSD4 | MP-SHCN $\hspace{0.02em}$  | 28.96$\pm$0.13
>
> +++++++++++++++++++++++++++++++++++
>
> $\bf{Table}$ $\bf{10}$: Comparison of RMSE of ZFC-SHCN to its two variants (i.e., with different topological summaries) on COVID-19 in TX.
>
> +++++++++++++++++++++++++++++++++++
>
> Dataset $\hspace{2em}$ | Architecture | RMSE
>
> +++++++++++++++++++++++++++++++++++
>
> TX: COVID-19  | ZFC-SHCN $\hspace{0.55em}$ | $^{**}$27.81$\pm$1.72
>
> +++++++++++++++++++++++++++++++++++
>
> TX: COVID-19  | DSF-SHCN $\hspace{0.55em}$ | 28.95$\pm$1.83
>
> +++++++++++++++++++++++++++++++++++
>
> TX: COVID-19  | MP-SHCN $\hspace{0.55em}$ | 28.12$\pm$1.80
>
> +++++++++++++++++++++++++++++++++++
>
> [2] Chen, Yuzhou, et al. "TAMP-S2GCNets: coupling time-aware multipersistence knowledge representation with spatio-supra graph convolutional networks for time-series forecasting." ICLR 2022.
>
>
> Q4: ``The experimental results, especially in Table 3, show that the proposed method seems to create different models for different evaluation indicators for a single problem. The same model should be used to compare different indicators in the evaluation. If you want to show that each indicator can be set appropriately, you should compare them separately.''
>
> $\bf{A}$: Table 3 and others compare the same model ZFC-SHCN using different indicators across various tasks and datasets. No different models for different evaluation indicators for a single problem are created.

---

> ### Author Response · Authors · 2022-07-31
> **Response to Technical Questions (2/3)**
>
> Q2: ``The proposed method introduces zigzag filtration features and supra-Hodge convolution features in addition to the conventional graph convolution features. Although the overall framework has been evaluated, it is not clear which features have what effect on each of them. Since it is not possible to determine whether all of the elements are necessary, I wonder if a comparison with a system that excludes any of the information would make the observation possible. Are such comparisons being made?''
>
> $\bf{A}$: We agree with this comment. We originally designed three variants of the ZFC-SHCN model and reported the results of ablation study on two datasets (i.e., PeMSD4 and COVID-19 in TX) in the paper (see Table 4 in Section 5.5). Now, to emphasize the importance of different components in ZFC-SHCN, we conducted additional experiments/ablation studies on PeMSD8, COVID-19 in CA. and COVID-19 in PA. As Tables 6, 7, and 8 show, we obtain similar conclusions compared to Table 4 in Section 5,  that is, our ZFC-SHCN significantly outperforms its three variants on PeMSD8, COVID-19 in CA, and COVID-19 in PA. In view of these findings, we can safely conclude that all three components - zigzag filtration features, supra-Hodge convolution operation, and graph convolution operation are important components for spatio-temporal forecasting tasks.
>
> $\bf{Table}$ $\bf{6}$: Ablation study on PeMSD8.
>
> +++++++++++++++++++++++++++++++++++++++++++++++++++++++++++++++++++++
>
> Dataset | Architecture $\hspace{16.5em}$| RMSE
>
> +++++++++++++++++++++++++++++++++++++++++++++++++++++++++++++++++++++
>
> PeMSD8 | ZFC-SHCN $\hspace{16.8em}$ | $^{***}$21.83$\pm$0.15
>
> +++++++++++++++++++++++++++++++++++++++++++++++++++++++++++++++++++++
>
> PeMSD8 | ZFC-SHCN W/o Graph convolution in spatial dimension  | 22.36$\pm$0.17
>
> +++++++++++++++++++++++++++++++++++++++++++++++++++++++++++++++++++++
>
> PeMSD8 | ZFC-SHCN W/o ZFC convolution  $\hspace{9em}$| 22.30$\pm$0.21
>
> +++++++++++++++++++++++++++++++++++++++++++++++++++++++++++++++++++++
>
> PeMSD8 | ZFC-SHCN W/o Supra-Hodge convolution  $\hspace{5.3em}$| 23.07$\pm$0.19
>
> +++++++++++++++++++++++++++++++++++++++++++++++++++++++++++++++++++++
>
>
>
> $\bf{Table}$ $\bf{7}$: Ablation study on COVID-19 in CA.
>
> ++++++++++++++++++++++++++++++++++++++++++++++++++++++++++++++++++++++++++
>
> Dataset $\hspace{2em}$ | Architecture $\hspace{16em}$| RMSE
>
> ++++++++++++++++++++++++++++++++++++++++++++++++++++++++++++++++++++++++++
>
> CA: COVID-19 | ZFC-SHCN $\hspace{16.8em}$ | $^{*}$343.92$\pm$3.66
>
> ++++++++++++++++++++++++++++++++++++++++++++++++++++++++++++++++++++++++++
>
> CA: COVID-19 | ZFC-SHCN W/o Graph convolution in spatial dimension  | 347.95$\pm$3.97
>
> ++++++++++++++++++++++++++++++++++++++++++++++++++++++++++++++++++++++++++
>
> CA: COVID-19 | ZFC-SHCN W/o ZFC convolution  $\hspace{9em}$| 346.58$\pm$3.30
>
> ++++++++++++++++++++++++++++++++++++++++++++++++++++++++++++++++++++++++++
>
> CA: COVID-19 | ZFC-SHCN W/o Supra-Hodge convolution  $\hspace{5.3em}$| 347.61$\pm$5.51
>
> ++++++++++++++++++++++++++++++++++++++++++++++++++++++++++++++++++++++++++
>
> $\bf{Table}$ $\bf{8}$: Ablation study on COVID-19 in PA.
>
> ++++++++++++++++++++++++++++++++++++++++++++++++++++++++++++++++++++++++++
>
> Dataset $\hspace{2em}$ | Architecture $\hspace{16.em}$| RMSE
>
> ++++++++++++++++++++++++++++++++++++++++++++++++++++++++++++++++++++++++++
>
> PA: COVID-19 | ZFC-SHCN $\hspace{16.8em}$ | $^{**}$101.65$\pm$1.53
>
> ++++++++++++++++++++++++++++++++++++++++++++++++++++++++++++++++++++++++++
>
> PA: COVID-19 | ZFC-SHCN W/o Graph convolution in spatial dimension  | 106.21$\pm$1.99
>
> ++++++++++++++++++++++++++++++++++++++++++++++++++++++++++++++++++++++++++
>
> PA: COVID-19 | ZFC-SHCN W/o ZFC convolution  $\hspace{9em}$| 102.82$\pm$1.34
>
> ++++++++++++++++++++++++++++++++++++++++++++++++++++++++++++++++++++++++++
>
> PA: COVID-19 | ZFC-SHCN W/o Supra-Hodge convolution  $\hspace{5.3em}$| 103.53$\pm$2.30
>
> ++++++++++++++++++++++++++++++++++++++++++++++++++++++++++++++++++++++++++

---

> ### Author Response · Authors · 2022-07-31
> **Response to Technical Questions (1/3)**
>
> Thank you very much for appreciating the importance and novelty of our ideas on zigzag filtration as the key time-aware topological information and mathematical foundations behind them! We are also particularly grateful for the thought-provoking question on other possibilities to summarize time-aware topological information which motivated us to explore some interesting extensions of ZFC. We have addressed your questions below and also added the respective updates into the paper (see Remark E.1 (Q1) in Appendix E, Remark E.3 (Q2) in Appendix E, Remark E.4 (Q3) in Appendix E). We would be grateful for any additional suggestions on the experimental or methodological validation that could further convince you in the ZFC-SHCN capabilities and could boost your assessment of ZFC-SHCN.
>
> Q1: ``Although the authors have compared the proposed method with conventional methods that do not include topology information, they have not examined whether the proposed method is appropriate as a method that takes topology information into account.''
>
> $\bf{A}$: Thank you very much for bringing this up! To examine whether our proposed zigzag filtration curve (ZFC) is appropriate for spatio-temporal forecasting tasks, for fair comparison, we now also perform ZFC-SHCN and ZPI-SHCN (i.e., replacing zigzag filtration curve with zigzag persistence image (ZPI) ) on COVID-19 (TX) and PeMSD4 datasets. Tables 4 and 5 show that the architecture based on zigzag filtration curve (i.e., ZFC-SHCN) significantly outperforms than the same architecture based on zigzag persistence image (i.e., ZPI-SHCN), on both datasets. These findings might be explained by the fact that, especially for small- and medium-size datasets, ZFC tends to better capture the topological signal than ZPI which, being a 2-d representation, might be more noisy.
>
> Note that, we also perform a one-sided two-sample $t$-test between the best result and the best performance achieved by the runner-up, where $*, **, ***$ denote $p$-value $< 0.1, 0.05, 0.01$ (i.e., denote significant, statistically significant, highly statistically significant results, respectively.
>
> $\bf{Table}$ $\bf{4}$: Comparison between ZFC and ZPI on COVID-19 in TX.
>
> +++++++++++++++++++++++++++++++++++++++
>
> Dataset $\hspace{2.2em}$| Architecture | RMSE
>
> +++++++++++++++++++++++++++++++++++++++
>
> TX: COVID-19 | ZFC-SHCN $\hspace{0.6em}$ | $^{**}$27.81$\pm$1.72
>
> +++++++++++++++++++++++++++++++++++++++
>
> TX: COVID-19 | ZPI-SHCN $\hspace{0.6em}$  | 29.26$\pm$1.86
>
> +++++++++++++++++++++++++++++++++++++++
>
>
> $\bf{Table}$ $\bf{5}$: Comparison between ZFC and ZPI  on PeMSD4.
>
> +++++++++++++++++++++++++++++++++++++++
>
> Dataset | Architecture | RMSE
>
> +++++++++++++++++++++++++++++++++++++++
>
> PeMSD4 | ZFC-SHCN $\hspace{0.5em}$| $^{***}$28.58$\pm$0.15
>
> +++++++++++++++++++++++++++++++++++++++
>
> PeMSD4 | ZPI-SHCN $\hspace{0.5em}$| 29.00$\pm$0.17
>
> +++++++++++++++++++++++++++++++++++++++

---

> ### Comment · Reviewer_isoH · 2022-08-03
> **Thanks for the answer.**
>
> A1&A3: Thank you for the new and meaningful information. It is impossible to cover everything, so these information should be sufficient for this problem. However, there are already various results of graph analysis using TDA, so it would be good to mention them even briefly in the Related work or somewhere else.
>
>
> A2: Thank you for the new verification and information. I think it is very interesting and useful information that each function has some meaning. It would be useful for users to clarify under what conditions each function is effective, so we look forward to more in-depth discussions in the future.
>
> A4: I understand.However, as Reviewer vJan pointed out, there are some parts that are easily misunderstood by readers, and I recommend a review from that point of view again.
>
> Overall, I think the value of the proposed method is clear. We will revise our evaluation, but please continue to refine it.

---

> > ### Author Response · Authors · 2022-08-03
> > **Thank you very much and additional details (2/2)**
> >
> > Regarding the choice of simplicial complexes, we have also conducted additional experiments between (i) ZFC-SH$^1$CN based on Supra-Hodge $1$-Laplacian (i.e., utilizing information of 0-simplices, 1-simplices, and 2-simplices) and (ii) ZFC-SH$^2$CN based on Supra-Hodge $1$-Laplacian (i.e., utilizing information of 1-simplices, 2-simplices, and 3-simplices) on PeMSD4 and COVID-19 in TX dataset (see Tables 13 and 14 in our response to Reviewer vJan (Q6) and Remark F.3 in Appendix F)). Experimental results show that ZFC-SH$^1$CN consistently outperforms ZFC-SH$^2$CN. We tend to attribute such findings to the fact that we observe a very low number of 3-simplices. For instance, in PeMSD4 at each timestamp we observe around $\bf{29}$ 2-simplices (filled triangles) but we observe only around $\bf{2}$ 3-simplices (tetrahedron). Furthermore, it is worth to mention that: for PeMSD4 dataset, ZFC-SH$^2$CN is still very competitive compared to other baselines (i.e., slightly worse than the runner-up - Z-GCNETs); for COVID-19 in TX, the performance of ZFC-SH$^2$CN is much better the runner-up (i.e., Z-GCNETs; in RMSE: ZFC-SH$^2$CN - 29.39$\pm$1.93 $\bf{vs.}$ Z-GCNETs - 46.04$\pm$1.78). Nevertheless, we believe that higher order simplices, if observed in the targeted network, has a potential to boost the model performance.
> >
> > $\bf{Table}$ $\bf{13}$: Comparison between ZFC-SH$^1$CN vs ZFC-SH$^2$CN on PeMSD4.
> >
> > ++++++++++++++++++++++++++++++++++++++++++
> >
> > Dataset | Architecture | RMSE
> >
> > ++++++++++++++++++++++++++++++++++++++++++
> >
> > PeMSD4 | ZFC-SH$^1$CN | $^{***}$28.58$\pm$0.15
> >
> > ++++++++++++++++++++++++++++++++++++++++++
> >
> > PeMSD4 | ZFC-SH$^2$CN | 29.15$\pm$0.30
> >
> > ++++++++++++++++++++++++++++++++++++++++++
> >
> >
> > $\bf{Table}$ $\bf{14}$: Comparison between ZFC-SH$^1$CN vs ZFC-SH$^2$CN on COVID-19 in TX.
> >
> > ++++++++++++++++++++++++++++++++++++++++++
> >
> > Dataset $\hspace{2em}$ | Architecture | RMSE
> >
> > ++++++++++++++++++++++++++++++++++++++++++
> >
> > TX: COVID-19 | ZFC-SH$^1$CN | $^{**}$27.81$\pm$1.72
> >
> > ++++++++++++++++++++++++++++++++++++++++++
> >
> > TX: COVID-19 | ZFC-SH$^2$CN | 29.39$\pm$1.93
> >
> > ++++++++++++++++++++++++++++++++++++++++++
> >
> > We will report more in-depth discussions about the choices of topological summary based on the zigzag filtration function and simplicial complexes in the final version.
> >
> > Q7: ``As Reviewer vJan (R4) pointed out, there are some parts that are easily misunderstood by readers, and I recommend a review from that point of view again.''
> >
> > $\bf{A}$:  We agree with this comment and we will go over the paper to enhance its clarity.

---

> > > ### Comment · Reviewer_isoH · 2022-08-04
> > > **Thanks for the discussion**
> > >
> > > Thanks for the additional verification.  I hope the final version will be better.

---

> > ### Author Response · Authors · 2022-08-03
> > **Thank you very much and additional details (1/2)**
> >
> > Thank you very much for appreciating the novelty and potential of ZFC, for the interactive discussion and for raising the score!
> >
> > Q5: ``There are already various results of graph analysis using TDA, so it would be good to mention them even briefly in the Related work or somewhere else.''
> >
> > $\bf{A}$: We agree, it’s indeed an actively developing area which deserves its own overview. We think that the best option will be to add a separate subsection on TDA for graph learning into the related work and we will do it in the final version.
> >
> > Q6: ``I think it is very interesting and useful information that each function has some meaning. It would be useful for users to clarify under what conditions each function is effective, so we look forward to more in-depth discussions in the future.''
> >
> > $\bf{A}$: Thanks very much for this suggestion!
> > In short, we believe that the ZFC component is of particular use for sparse heterogeneous networks with richer topology, while Supra-Hodge is of particular utility when the network is characterized by distinct multi-node interactions and, hence, describing diffusion across higher order substructures becomes essential. For example, family units can be modeled as simplices, and accounting for diffusion across such higher order substructures plays an important role in tracking propagation of infectious disease agents or information on social networks. However, for more homogenous networks the contribution of ZFC and Supra-Hodge Laplacians might be limited if any. Below, we provide a longer answer with experimental results.
> >
> > To clarify the advantages of zigzag filtration curve (ZFC) over the zigzag persistence image (ZPI), we have conducted an additional comparison experiment between ZFC and ZPI on traffic dataset (i.e., PeMSD4) and COVID-19 in TX dataset (see Tables 1 and 2 in our response to Reviewer m8mW (Q1) and Remark E.1 in Appendix E)). Note that, for fair comparison, based on the existing model architecture (i.e., our ZFC-SHCN model), we replace ZFC with ZPI - thus we can have the baseline, i.e., ZPI-SHCN. The experimental results show that the ZFC representation (i.e., ZFC-SHCN) significantly outperforms ZPI-SHCN on both datasets. These findings might be explained by the fact that, especially for small- and medium-size datasets, ZFC tends to better capture the topological signal than ZPI which, being a 2-d representation, might be more noisy. Furthermore, another advantage of ZFC is computational efficiency ($\bf{these}$ $\bf{results}$ $\bf{are}$ $\bf{new}$ $\bf{and}$ $\bf{have}$ $\bf{not}$ $\bf{be}$ $\bf{reported}$ $\bf{before}$). For instance, for PeMSD4: the generation of ZFC and average training time per epoch of ZFC-SHCN are $\bf{0.52 s}$ and $\bf{27.53 s}$; the generation of ZPI and average training time per epoch of ZPI-SHCN are $\bf{0.86 s}$ and $\bf{32.57 s}$.
> >
> > ​​$\bf{Table}$ $\bf{1}$: Ablation study on COVID-19 in TX.
> >
> > ++++++++++++++++++++++++++++++++++++++++++
> >
> > Dataset $\hspace{2em}$| Architecture | RMSE
> >
> > ++++++++++++++++++++++++++++++++++++++++++
> >
> > TX: COVID-19 | ZFC-SHCN $\hspace{0.3em}$ | $^{**}$27.81$\pm$1.72
> >
> > ++++++++++++++++++++++++++++++++++++++++++
> >
> > TX: COVID-19 | ZPI-SHCN $\hspace{0.3em}$ | 29.26$\pm$1.86
> >
> > ++++++++++++++++++++++++++++++++++++++++++
> >
> > TX: COVID-19 | ZFC-GCN $\hspace{0.4em}$ | 28.71$\pm$1.90
> >
> > ++++++++++++++++++++++++++++++++++++++++++
> >
> >
> >
> > $\bf{Table}$ $\bf{2}$: Ablation study on PeMSD4.
> >
> > ++++++++++++++++++++++++++++++++++++++++++
> >
> > Dataset | Architecture | RMSE
> >
> > ++++++++++++++++++++++++++++++++++++++++++
> >
> > PeMSD4 | ZFC-SHCN | $^{***}$28.58$\pm$0.15
> >
> > ++++++++++++++++++++++++++++++++++++++++++
> >
> > PeMSD4 | ZPI-SHCN | 29.00$\pm$0.17
> >
> > ++++++++++++++++++++++++++++++++++++++++++
> >
> > PeMSD4 | ZFC-GCN | 28.90$\pm$0.26
> >
> > ++++++++++++++++++++++++++++++++++++++++++
> >
> > (see the choice of simplicial complexes in ``Thank you very much and additional details (2/2)'')

---

> ### Author Response · Authors · 2022-08-09
> **Thank you very much for the inspiring feedback!**
>
> We have added a subsection on TDA for graph learning in the related work (see "Topological Data Analysis for Graph Learning" (from lines#81-92) in Related Work)." (note that, due to space limitation, we move part of experimental settings to Remark E.6 in Appendix).

---

### Official Review · Reviewer_u6Ln · 2022-07-12

**Rating:** 7
**Confidence:** 3
**Soundness:** 3 good
**Presentation:** 3 good
**Contribution:** 2 fair

**Summary:**

This paper addresses the problem of  capturing information other than pairwise relations in GNN. A time-conditioned topological knowledge representation is proposed, and related theoretical properties are presented. The topic is very interesting and influential.

**Questions:**

The complexity should be discussed and some other type of tasks should be done.

**Limitations:**

No suggestions.

**Strengths And Weaknesses:**

Strengths
- good idea.
- there is theoretical stability guarantees.

Weaknesses
- the complexity should be discussed.
- it is not very convincing that the expression is enough for various of tasks.

---

> ### Author Response · Authors · 2022-07-31
> **Response to Technical Questions**
>
> Thank you very much for recognizing the novelty of the proposed ideas and the importance of the topic and for the thought provoking question on diversifying the tasks! We have addressed your questions below and also added the respective updates into the paper (see Remark F.1 (Q1) in Appendix F, Remark E.2 (Q2) in Appendix E). We would be grateful for any additional suggestions on the experimental or methodological validation that could further convince you in the ZFC-SHCN capabilities and could boost your assessment of ZFC-SHCN.
>
> Q1: ``The complexity should be discussed.''
>
> $\bf{A}$: The computational complexity of the overall approach is: $\mathcal{O}(N^2 + \Upsilon^{\delta} + \Xi\_{k} \omega F\_{k} d\_{out} + \Xi\_{k} \omega^2 d\_{out}/2 + d\_{out} \sum^{t-1}\_{\ell = t- \omega}\Xi^{(\ell)}\_{k+1} + W\_{GRU})$, including (i) graph convolution in spatial dimension: $\mathcal{O}(N^2)$ (where $N$ is the number of nodes), (ii) zigzag filtration curve (ZFC): $\mathcal{O}(\Upsilon^{\delta})$ (where $\Upsilon$ represents the number of points in time interval and $\delta \in [2, 2.373)$), (iii) supra-Hodge convolution in temporal dimension: $\mathcal{O}(\Xi\_{k} \omega F\_{k} d\_{out} + \Xi\_{k} \omega^2 d\_{out}/2 + d\_{out} \sum^{t-1}\_{\ell = t- \omega}\Xi^{(\ell)}\_{k+1})$ (where $F\_{k}$ is the number of $k$-simplex attribute features, $\omega$ is the sliding window size, $d\_{out}$ is the output dimension of the supra-Hodge convolution layer, and $\Xi^{(\ell)}\_{k+1}$ is the number of $(k+1)$-simplex $\Xi\_{k+1}$ at the $\ell$-th layer). Note that, in our paper, we denote the number of nodes as $N$ (i.e., $\Xi\_0$), the number of edges as $M$ (i.e., $\Xi\_1$), and the number of filled triangles as $Q$ (i.e., $\Xi\_2$), (v) GRU: $\mathcal{O}(W\_{GRU})$ (where $W\_{GRU}$ is the total number of parameters in the GRU). Furthermore, in Section 5.6 (the main body), we have provided the computational complexities of incidence matrices $\boldsymbol{B}\_1$ and $\boldsymbol{B}\_2$ are $\mathcal{O}(N + M)$ and $\mathcal{O}(M + Q)$, respectively.
>
> Q2: ``It is not very convincing that the expression is enough for various of tasks.''
>
> $\bf{A}$: Thank you for raising this question which has motivated us to look at additional experiments and strengthen our paper.  In our original submission, we validated our ZFC-SHCN model on five diverse data types for the node values prediction in spatio-temporal graphs, i.e., (i) COVID-19 datasets (CA, PA, and TX), (ii) traffic datasets (PeMSD4 and PeMSD8), (iii) synthetic multivariate time-series (MTS) datasets based on 268 vector autoregression (VAR), (iv) surface air temperature (CA, PA, and TX), and (v) Ethereum blockchain network (Bytom). As suggested, to assess the utility of our approach for other tasks, we now also conducted spatio-temporal forecasting for graph values (i.e., on graph-level) on the wind energy dataset [1]. In this experiment, we use hourly wind speed data to predict the total wind speed from 57 stations in East Coast including Massachusetts, Connecticut, New York, and New Hampshire. The evaluation results are summarized in Table 3. We find that our ZFC-SHCN outperforms the next best competitor, with up to 39.20\% of improvement, and the results are highly statistically significant. Therefore, we believe that the highly competitive performances of ZFC-SHCN for both node values and graph value forecasting in spatio-temporal networks, makes the expression sufficiently convincing for a broad range of forecasting tasks. In the future, we will also explore the utility of ZFC-SHCN for inductive link prediction.
>
> Note that, we also perform a one-sided two-sample $t$-test between the best result and the best performance achieved by the runner-up, where $*, **, ***$ denote $p$-value $< 0.1, 0.05, 0.01$ (i.e., denote significant, statistically significant, highly statistically significant results, respectively.
>
> $\bf{Table}$ $\bf{3}$: Forecasting results of ZFC-SHCH and baselines on wind energy dataset.
>
> +++++++++++++++++++++++++++++++++++++++++++
>
> Dataset $\hspace{1.8em}$ | $\hspace{0.5em}$ Model $\hspace{0.5em}$ | RMSE
>
> +++++++++++++++++++++++++++++++++++++++++++
>
> Wind Energy | ZFC-SHCN | $^{***}$1.07$\pm$0.02
>
> +++++++++++++++++++++++++++++++++++++++++++
>
> Wind Energy | ZPI-SHCN | 1.30$\pm$0.02
>
> +++++++++++++++++++++++++++++++++++++++++++
>
> Wind Energy | Z-GCNETs | 1.68$\pm$0.05
>
> +++++++++++++++++++++++++++++++++++++++++++
>
> Wind Energy | $\hspace{0.1em}$ AGCRN $\hspace{0.5em}$ | 1.52$\pm$0.03
>
> +++++++++++++++++++++++++++++++++++++++++++
>
> Wind Energy | StemGNN| 1.76$\pm$0.03
>
> +++++++++++++++++++++++++++++++++++++++++++
>
> [1] Ghaderi, Amir, Borhan M. Sanandaji, and Faezeh Ghaderi. "Deep forecast: Deep learning-based spatio-temporal forecasting." ICML 2017 Time Series Workshop.

---

> ### Author Response · Authors · 2022-08-01
> **Thank you very much!**
>
> Thank you very much for raising the score and for recognizing the novelty of the ZFC approach!

---

### Official Review · Reviewer_m8mW · 2022-07-13

**Rating:** 6
**Confidence:** 4
**Soundness:** 4 excellent
**Presentation:** 4 excellent
**Contribution:** 2 fair

**Summary:**

The paper proposes a new time-aware GNN networks called Zigzag Filtration Curve based Supra-Hodge Convolution Networks(ZFC-SHCN). Specifically, the networks composes 1) the simplicial neural networks with an efficient zigzag persistence curve and 2) a new temporal multiplex graph representation module. In practice, the experimental results on several synthetic and real world datasets demonstrate the effectiveness and efficiency of the proposed ZFC-SHCN.

**Questions:**

Same as weaknesses.

**Strengths And Weaknesses:**

Strengths

+ The paper is well-written and well-organized.
+ The idea, amplifying the power of SNNs with a time-conditioned topological knowledge representation in a form of zigzag persistence, is interesting and novel.
+ The experimental results show ZFC-SHCN outperforms other baselines.
+ Codes are provided.

Weaknesses

+ It worth noting that the gains from ZFC convolution and Supra-Hodge convolution in Tabel 4 are not significant as expected, It would be fairer if the authors could provide more comparison experiments. E.g.  Replacing ZFC with zigzag persistence image
(ZPI) to show the gain from ZFC, replacing Supra-Hodge convolution with $GCN_{temp}$ to show the gain from Supra-Hodge convolution, etc. Or theoretical analysis of the differences is given.

---

> ### Author Response · Authors · 2022-07-31
> **Response to Technical Questions**
>
> Thank you very much for appreciating the novelty of  the time-conditioned topological knowledge representation in a form of zigzag persistence and suggesting the additional ablation experiments that have emphasized the ZFC-SHCN capabilities! We have addressed your questions below and also added the respective updates into the paper (see Remark E.1 in Appendix E). We would be grateful for any additional suggestions on the experimental or methodological validation that could further convince you in the ZFC-SHCN capabilities and could boost your assessment of ZFC-SHCN.
>
> Q1: ``It worth noting that the gains from ZFC convolution and Supra-Hodge convolution in Table 4 are not significant as expected, It would be fairer if the authors could provide more comparison experiments. E.g. Replacing ZFC with zigzag persistence image (ZPI) to show the gain from ZFC, replacing Supra-Hodge convolution with $GCN_{temp}$ to show the gain from Supra-Hodge convolution, etc. Or theoretical analysis of the differences is given.''
>
> $\bf{A}$: We agree with this comment. Following the Reviewer’s suggestion, we conducted the additional experiments (i) replacing ZFC with zigzag persistence image (ZPI), i.e., ZPI-SHCN and (ii) replacing Supra-Hodge convolution with GCN$_{temp}$, i.e., ZFC-GCN, on both PeMDS4 and COVID-19 (in Texas state) datasets. First, Tables 1 and 2 demonstrate that for both datasets, the ZFC representation (i.e., ZFC-SHCN) significantly outperforms zigzag persistence image (i.e., ZPI-SHCN). These findings might be explained by the fact that, especially for small- and medium-size datasets, ZFC tends to better capture the topological signal than ZPI which, being a 2-d representation, might be more noisy. Second, Tables 1 and 2 indicate that compared to ZFC-GCN (which uses graph convolution operation to capture nodes’ information), ZFC-SHCN always achieves better forecasting performance. We attribute these findings to the fact that supra-Hodge convolution operation integrates knowledge on important interactions among higher-order structures (i.e., edges, filled triangles - beyond the node-space) into graph learning.
> For COVID-19 (in TX) dataset, the relative gain of ZFC-SHCN over the runner-up (i.e., ZFC-GCN) in RMSE is 3.13\%; for PeMSD4 dataset, the relative gain of ZFC-SHCN over the runner-up (i.e., ZFC-GCN) in RMSE is 1.11\%.
> We also perform a one-sided two-sample $t$-test between the best result and the best performance achieved by the runner-up, where $\*$, $\**$, $\***$ denote $p$-value $< 0.1, 0.05, 0.01$ (i.e., denote significant, statistically significant, highly statistically significant results, respectively.
>
> $\bf{Table}$ $\bf{1}$: Ablation study on COVID-19 in TX.
>
> ++++++++++++++++++++++++++++++++++++++++++
>
> Dataset $\hspace{2em}$| Architecture | RMSE
>
> ++++++++++++++++++++++++++++++++++++++++++
>
> TX: COVID-19 | ZFC-SHCN $\hspace{0.3em}$ | $^{**}$27.81$\pm$1.72
>
> ++++++++++++++++++++++++++++++++++++++++++
>
> TX: COVID-19 | ZPI-SHCN $\hspace{0.3em}$ | 29.26$\pm$1.86
>
> ++++++++++++++++++++++++++++++++++++++++++
>
> TX: COVID-19 | ZFC-GCN $\hspace{0.4em}$ | 28.71$\pm$1.90
>
> ++++++++++++++++++++++++++++++++++++++++++
>
>
>
> $\bf{Table}$ $\bf{2}$: Ablation study on PeMSD4.
>
> ++++++++++++++++++++++++++++++++++++++++++
>
> Dataset | Architecture | RMSE
>
> ++++++++++++++++++++++++++++++++++++++++++
>
> PeMSD4 | ZFC-SHCN | $^{***}$28.58$\pm$0.15
>
> ++++++++++++++++++++++++++++++++++++++++++
>
> PeMSD4 | ZPI-SHCN | 29.00$\pm$0.17
>
> ++++++++++++++++++++++++++++++++++++++++++
>
> PeMSD4 | ZFC-GCN | 28.90$\pm$0.26
>
> ++++++++++++++++++++++++++++++++++++++++++

---

> > ### Comment · Reviewer_m8mW · 2022-08-07
> > **Responses to authors. Raise my score.**
> >
> > Thanks for the authors' well-written rebuttal and detailed experimental results. The responses almost address all my questions and concerns. Therefore, I decide to raise my score from 5 to 6 (weak accept).

---

> > > ### Author Response · Authors · 2022-08-07
> > > **Thank you for reading our response!**
> > >
> > > Thanks a lot for your effort in reviewing our paper and re-evaluating it. We are very grateful for the constructive and inspiring feedback and of course for raising the score!

---

### Author Response · Authors · 2022-07-31
**General Response**

We would like to thank the Reviewers for very constructive, detailed and motivating feedback, and for recognizing the novelty and potential of the proposed ideas on zigzag filtration and Supra-Hodge convolution in conjunction with time series forecasting. We have uploaded the Rebuttal Revisions for both main paper and supplementary material. Below we provide a detailed response to each of the Reviewers’ questions.

---

### Meta-Review · Area_Chair_G8iH · 2022-08-27

**Recommendation:** Accept
**Confidence:** Certain

**Metareview:**

Their new ideas are highly appreciated by the authors. The authors' enthusiastic revision of the paper is also good.
It is also nice that the benefits are clear and the advantages are clearly presented both theoretically and experimentally.

**Award:**

No

---

### Decision · Program_Chairs · 2022-09-14

Accept